

# Dynamic INtegrated Gap-filling and partitioning for OzFlux (DINGO)

5  Jason Beringer[1], Ian McHugh[2], Lindsay B. Hutley[3], Peter Isaac[4] and Natascha Kljun[5]

[1]School of Earth and Environment (SEE), The University of Western Australia, Crawley, WA, 6009, Australia.

[2]School of Earth, Atmosphere and Environment, Monash University, Clayton, 3800, Australia.

[3]School of Environment, Research Institute for the Environment and Livelihoods, Charles Darwin University, NT, Australia 0909.

10  [4]School of Earth, Atmosphere and Environment, Monash University, Clayton, 3800, Australia.

[5]Department of Geography, Swansea University, Singleton Park, Swansea, Wales SA2 8PP, United Kingdom.

*Corresponding to: Jason Beringer, tel. +61 409355496, e-mail: Jason.Beringer@uwa.edu.au*




## Abstract

Standardised, quality-controlled and robust data from flux networks underpin the understanding of ecosystem processes and tools necessary to support the management of natural resources including water, carbon and nutrients for environmental and production benefits. The Australian regional flux network (OzFlux) currently has 23 active sites and aims to provide a

continental-scale national research facility to monitor and assess Australia's terrestrial biosphere and climate for improved predictions. Given the need for standardised and effective data processing of flux data we have developed a software suite called the Dynamic INtegrated Gap-filling and partitioning for OzFlux (DINGO) that enables gap-filling and partitioning of the primary fluxes into ecosystem respiration and gross primary productivity and subsequently provides diagnostics and results. We outline the processing pathways and methodologies that are applied in DINGO (v12a) to OzFlux data including 1) gap-

filling of meteorological and other drivers; 2) gap-filling of fluxes using artificial neural networks; 3) the u* threshold determination; 4) partitioning into ecosystem respiration and gross primary productivity; and 5) diagnostic, summary and results outputs. Opportunities remain for DINGO to incorporate robust measurements of uncertainty for application in land management and carbon accounting. In addition, footprint information is crucial in understanding and interpreting the scale and spatial influence of flux measurements. Both these features are scheduled for the next release (v13) but are detailed here.

DINGO was developed for Australian data but the framework is applicable to any flux data or regional network. Quality data from robust systems like DINGO ensure the utility and uptake of the flux data and facilitates synergies between flux, remote sensing and modelling.





# 1 Introduction

OzFlux is the regional Australian and New Zealand flux tower network that aims to provide a continental-scale national research facility to monitor and assess Australia's terrestrial biosphere and climate for improved predictions (Beringer et al., 2016). High quality and reliable data are a crucial foundation in achieving the objectives of the OzFlux network (Beringer et al., 2016) and underpin the process understanding needed to: 1) support sound management of natural resources including water, carbon and nutrient resources for environmental and production benefits; 2) monitor, assess, predict and respond to climate change and variability; 3) improve weather and environmental information and prediction; 4) support disaster management and early warning systems needed to meet Australia's priorities in national security; and 5) ensure that Earth system models used to underpin Australia's policies and commitments to international treaties adequately represent Australian terrestrial ecosystem processes (Beringer et al., 2016).

Beringer et al. (2016) provide an overview of the evolution, design and current status of OzFlux as well as a brief summary of the instrumentation and data collection that forms the backbone of the network. A detailed description of the quality control and post-processing of eddy covariance data using OzFluxQC and the data pathway to curation is given by Isaac et al. (2016). In summary from Beringer et al. (2016), most sites have data loggers that provide the average (usually over 30 minutes) covariances that are processed through 6 levels using the OzFluxQC standard software processing scripts. Levels 1, 2 and 3 are as follows; L1 - the raw data as received from the flux tower, L2 - quality-controlled data, and L3 - post-processed, corrected but not gap-filled data. Quality control measures by OzFluxQC are applied at L2 and comprise checks for plausible value ranges, spike detection and removal, manual exclusion of date and time ranges and diagnostic checks for all quantities used in the flux correction calculations. The quality checks make use of the diagnostic information from the sonic anemometer and the infra-red gas analyser. For sites calculating fluxes from the averaged covariances, post-processing includes 2-dimensional coordinate rotation, low- and high-pass frequency correction, conversion of virtual heat flux to sensible heat flux and application of the WPL correction to the latent heat and $CO_2$ fluxes (see Burba (2013) for general description of the data processing pathways). Steps performed at L3 include the correction of the ground heat flux for storage in the layer above the heat flux plates (Mayocchi and Bristow, 1995) and correction of the $CO_2$ flux data for storage in the canopy (where available). OzFlux data are available at http://data.ozflux.org.au/. OzFlux sites submit their data to FLUXNET at L3.

Given the international need by the community for standardised data processing to enable effective comparison across biomes and to understand inter-annual variability (Papale et al., 2006), we have developed a software tool to address this need. In this paper we describe the development and testing of the Dynamic INtegrated Gap-filling and partitioning for Ozflux (DINGO) system that utilises the L3 data from OzFluxQC to gap-fill and partition the fluxes in ecosystem respiration (Fre) and gross primary productivity (GPP) and subsequently provides diagnostics and results. This paper is not intended to be a thorough review of the data processing but the application of standard techniques in DINGO for the flux community. DINGO is a




research version for OzFlux data whereas the OzFluxQC system (which has many similar features to DINGO) is considered an operational version. Quality data from robust systems like DINGO ensure the utility and uptake of the flux data and facilitates synergies between flux, remote sensing, modelling and canopy physiological studies. We conclude by looking ahead at the future direction of the DINGO system.

## 2 Approach

The overall approach used in DINGO is to take the L3 OzFluxQC data, which has gaps from data processing (data excluded due to values out of range, spike detection or manual exclusion of date and time ranges) or from site issues (instrument or power failure, herbivores, fire, eagles nests, cows, lightning, PI on sabbatical, etc.) and gap-fill and partition the data using a variety of data sources (Fig. 1). DINGO is programmed in python 2.7 and is currently at version 12a and publically available on GitHub (https://github.com/jberinge/DINGO12). It should be noted that v13 is scheduled to be released in July 2016 and will include uncertainty and footprint analysis and these are documented in sections 2.6 and 2.7, respectively. It is designed to work with OzFlux data produced in NetCDF format by the OzFluxQC (Isaac et al., 2016) and draws on Australian AWS data but could be adapted for other data sources across other flux sites. The primary interface for the user is through a text based control file that has information on site characteristics (name, latitude and longitude, the frequency of the flux measurements (30 or 60 minutes) and elevation), file paths (to the OzFluxQC NetCDF files and other ancillary data inputs), data processing options and data plotting and output formats. In general, prior to the processing steps below, any gaps in fluxes or meteorological quantities of less than two hours are filled by DINGO using linear interpolation. The pathway for processing is shown in Fig. 1 and each aspect is outlined below through each section.

## 2.1 Gap-filling of meteorological and other drivers

### 2.1.1 Gap-filling of meteorological drivers

In order to produce a continuous time series of fluxes and meteorological drivers for carbon accounting and other uses (Hutley et al., 2005) the first step is to generate a continuous time series of meteorological drivers that is in turn used for gap-filling of fluxes. In general, DINGO gap-fills missing meteorological data based on multiple data sources that include Bureau of Meteorology (BoM) operated automatic weather stations (AWS) and spatially gridded meteorological data at 0.1 degree resolution (Jones et al., 2009; Raupach et al., 2009). It then and chooses the 'best' source based on the best linear regression correlation with available site data. It uses BoM AWS data from nearby stations to create a constructed time series and then compares this with the same regression from gridded meteorological data. Given the relatively low density of AWS's across Australia, a station may not be representative of the flux tower location or may be too far away and therefore may not be well



correlated with local meteorology. In this case the correlation will be low and if the gridded data has a better correlation it will be used in preference. This procedure is repeated independently for each meteorological variable (Fig. 1). The general processing of AWS data is as follows:

1. search for the nearest BoM AWS stations and check to see if the station has data available for the same time period needed for the flux gap-filling and ensure that the AWS has the required meteorological variables,

2. obtain the 10 closest sites and extract the data accounting for different data formats. DINGO uses the Pandas library (http://pandas.pydata.org/) predominately for data manipulation. Here we do time stamp management and all processing is done based on local time. DINGO then looks for duplicates and or missing data and deals with 'NaNs' and QC flags from AWS data,

3. then correlate each AWS site with the available flux tower meteorology and use data from the best three out of ten sites tested,

4. the frequency at which to perform the correlation analysis between flux tower and AWS can be set to 'all' the available data or done by 'year' or 'month', and

5. then outputs and saves the 'best' AWS time series and creates plots.

The same process is repeated for the BoM spatially gridded data to obtain a correlation and linear regression with local meteorology. Gap-filling of the flux tower meteorological data then proceeds as follows:

1. the linear regression equation from the best available time series (based on the best correlation (AWS or gridded)) is used to adjust the time series to best match the flux site meteorology. The new 'correlated' variable is named {variable}_Corr (e.g. Ta_Corr). This variable is the best alternate meteorological time series adjusted by the linear regression with site meteorological data,

2. offset the best available time series by 30 minutes if it is from the BoM which has timestamps that are for the proceeding period, whereas flux tower dataloggers have timestamp at the end of the period,

3. missing meteorological data are then gap-filled using the {variable}_Corr time series and the new 'constructed' time series is saved as {variable}_Con (i.e. Ta_Con). An associated QC flag is generated {variable}_Con_QCFlag. If valid observations are present then they are used and the flag set to 1. If the data are gap-filled then flag set to 100, and

4. for each variable if at the end of all this there are still missing data then they are filled using a climatology approach using monthly averages of the diurnal time series for the variable and given a QC flag of 97.

As an example, for air temperature (Ta), we see the correlations of site Ta with the best nearby AWS (Fig. 2a) and the final gap-filled Ta series. This is repeated for absolute humidity (Ah), wind speed (WS), atmospheric pressure (P) and precipitation. The plots provide excellent diagnostics as the flux tower data is compared against the best available meteorology and this can detect anomalies such as instrumentation errors or processing errors. As an example in fig. 3 we see that a range threshold had been set incorrectly in OzFluxQC by the user and that accidently excluded data with an absolute humidity below 5 g m$^{-3}$.



## 2.1.2 Gap-filling of soil temperature and water content

The AWS data cannot be easily used to gap-fill soil temperature and moisture content. Therefore we use a process-based land surface model to simulate soil temperature and soil moisture. We use the BIOS2 model as described in Haverd et al. (2013a, 2013b) forced using remotely sensed estimates of leaf area index (LAI) and meteorology from the Bureau of Meteorology's

Australian Water Availability Project data set (BoM AWAP) (Jones et al., 2009; Raupach et al., 2009). BIOS2 is a fine-spatial-resolution (0.05°) offline modelling environment that includes a modification of the Community Atmosphere Biosphere Land Exchange land surface scheme (Wang et al., 2011) incorporating the Soil–Litter–Iso model (Haverd and Cuntz, 2010; Haverd et al., 2009) plus the Carnegie–Ames–Stanford Approach with Carbon–Nitrogen–Phosphorus (CASA–CNP) biogeochemical model (Wang et al., 2010). We calculate a regression equation of the modelled data versus the available site

data (Fig 4.) and then apply the regression equation to adjust the model time series. The modelled time series is then used for gap-filling and applied identically as for the meteorological drivers to produce gap-filled soil variables.

## 2.1.3 Use of Moderate-resolution imaging spectroradiometer (MODIS) data

MODIS satellite products provide necessary information for gap-filling of radiation (albedo and land surface temperature) and net ecosystem exchange (Fc) (using NDVI as a surrogate for GPP as described in section 2.3) as well as being valuable

information for site investigators for general use. The following process is undertaken:

1. MODIS cutouts for all available products (3 x 3 km around each tower) are extracted daily for all OzFlux sites using the SUDS (https://fedorahosted.org/suds/) python module to query the MODIS web service (https://daac.ornl.gov/MODIS/MODIS-menu/modis_webservice.html ),

2. the following products are extracted MOD09A1 (surface reflectance's), MOD13Q1 (NDVI and EVI), MOD15A2

(LAI/fPAR), MOD17A2 (GPP and Psnet), MOD16A2 (ET, PET, LE), MOD11A2 (day and night LST),

3. the MODIS products are filtered for anomalous values using the appropriate QC flags for each product. The QC tolerance can be set to 'normal' or 'tight' in the code, and

4. the 8 or 16 day products are interpolated to sub daily (variables XX_interp) using Scipy spline interpolation (order 1) function and then smoothed (variables xx_smooth) using a Savitzky-Golay filter to remove high frequency noise from

data (Savitzky and Golay, 1964).

An example of the final result for the enhanced vegetation index (EVI) is shown in Fig. 5.





## 2.2 Gap-filling of radiation variables

### 2.2.1 Incoming short wave radiation (insolation)

A two stage process is used to estimate half-hourly shortwave solar insolation ($F_{sd}$). First, Australia's Bureau of Meteorology (BOM) produces 5x5km spatially interpolated daily total insolation surfaces estimated from hourly visible-band geostationary

satellite data (currently MTSAT-1R) using a physical atmospheric radiative transfer model and water vapour amount estimated from a numerical weather prediction model (Grant et al., 2008). Data for the tile encompassing the location of the research site are used to provide estimates of daily total insolation. These data are regressed against site data and linearly transformed to correct any minor biases (coefficient of determination was routinely >0.9). Second, a temporal downscaling algorithm is applied to estimate insolation for the given measurement frequency of the site (generally half-hourly) from daily totals.

Insolation is calculated from application of Beer's Law, as follows:

$$F_{sd} = I_0 \cos Z \, e^{-km} \tag{1}$$

Where $I_o$ is direct beam top of atmosphere radiation, Z is zenith direction and k and m are atmospheric extinction coefficient and optical air mass term, respectively. $I_o$ is calculated using solar constant of 1367W m$^{-2}$ and the radius vector formulation of Duffie and Beckman (2103) to account for orbital eccentricity as follows:

$$I_0 = 1367.\left(1 + 0.034 \cos\left[\frac{360d}{365.25}\right]\right) \tag{2}$$

Where d is day of year. Zenith direction (Z) is derived from spherical geometry as per Oke (1987):

$$Z = \cos^{-1}(\sin\Phi \sin\delta + \cos\Phi \cos\delta \cos h) \tag{3}$$

Where $\Phi$ is latitude, $\delta$ solar declination (the angle between the solar beam and the equatorial plane) and $h$ the hour angle (the angular distance between the relevant longitude and that with which the sun is at 0º azimuth). Solar declination is calculated as follows (Cooper, 1969):

$$\delta = 23.4 \sin\left(\frac{360}{365} * [284 + d]\right) \tag{4}$$

The hour angle is (Oke, 1987):

$$h = 15(Sn - t) \tag{5}$$

Where Sn is solar noon and t is 24-hour time. Solar noon is calculated as:

$$Sn = 12 + 24\frac{\psi_{SM} - \psi_{LM}}{360} - EoT \tag{6}$$

Where $\psi_{SM}$ and $\psi_{LM}$ are the local standard time meridian and the local true longitude of the site, respectively (ºE), and EoT is the equation of time correction to account for the combined effects of orbital eccentricity and axial obliquity. EoT is calculated from the formulation of DiLaura (1984):

$$EoT = 0.17 \sin\left(4\pi\frac{d - 80}{373}\right) - \sin\left(2\pi\frac{d - 8}{355}\right) \tag{7}$$

Finally, the optical air mass term is calculated from Kasten and Young (1989):



$$m = \frac{e^{-0.0001184\,h}}{\cos Z + 0.50572(96.07995 - 180Z/\pi)^{-1.6364}}$$

(8)

Where h is the elevation above sea level (m). The extinction coefficient (k) is the only remaining unknown quantity in Eq. 1, and is optimised using the site observations. We first generated an upper envelope for daily insolation – taken to represent clear sky conditions – defined by selecting maxima for successive (non-overlapping) 14-day windows from the gap-filled daily time series (in the tropics it is necessary to exclude the wet season from this procedure, since substantial cloud mass routinely builds in the afternoons). The value of k is then optimised by minimising the sum of squares error between the envelope estimate and daily insolation totals calculated using eq. 1. The SciPy package of the Python programming language is used for the optimisation procedure.

Thus a clear sky estimate for each half-hourly period is obtained from the above procedure. Since no information about sub-daily variations in cloudiness is available, here it is assumed that cloudiness is constant throughout the day. As such, the clear sky half-hourly estimate for a given day is simply reduced by the ratio of the daily total BOM-estimated insolation to the daily total calculated clear sky insolation.

While k is likely to vary seasonally, here a single value is used for the sake of simplicity. By using site data to constrain estimates of  k, a more accurate diurnal course of $F_{sd}$ is obtained. This is crucial, because as the energy source for photosynthesis, insolation is routinely used as a driver in models that estimate daytime carbon fluxes. The calculation of theoretical site clear sky estimates of solar radiation also allows an estimate of cloudiness to be obtained. In turn, this is an important term in the determination of incoming long wave radiation, as described below.

**2.2.2 Incoming long wave radiation**

Similar to insolation, incoming long wave radiation ($F_{ld}$) is calculated in a two-step process. The first step is to estimate daily radiation totals. This s done by using the Stefan-Boltzmann relation to calculate daily average $F_{ld}$ from screen-level air temperature, as follows (Oke, 1987):

$$F_{ld} = \varepsilon \sigma T^4$$

(9)

Where $\varepsilon$ is emissivity,  $\sigma$ is the Stefan-Boltzmann constant (5.672 x $10^{-8}$W m$^{-2}$ K$^{-1}$) and T is temperature (K).  While $F_{ld}$ is received from all parts of the sky, measurements of water vapour (which influences $\varepsilon$) and temperature are generally only available at a discreet point near the surface. The clear sky formulation of Brutsaert  (1975) is used to take this into account by making approximating assumptions about average lapse rates of temperature and water vapour profiles based on surface values (Crawford and Duchon, 1999), and has been found to perform well under clear conditions (for example see Duarte et al. (2006)).  Its general form is as follows (Burman and Pochop, 1994):

$$\varepsilon = a \left( \frac{e_a}{T_a} \right)^b$$

(10)



Where $e_a$ is vapour pressure, $T_a$ is air temperature (K), and $a$ and $b$ are fitted parameters (equal to 1.24 and 0.143, respectively, in Brutsaert's original derivation for typical US conditions). Given the high emissivity of clouds relative to clear skies, cloudy conditions make estimation of ε more difficult. The approach adopted here follows that of Crawford and Duchon (1999),who further parameterised Brutsaert's equation to take account of cloud cover, as follows:

$$\varepsilon = clf + (1 - clf)\left(a\left(\frac{e_a}{T_a}\right)^b\right)$$

(11)

Where *clf* is the cloud fraction (ratio of cloudy to clear sky radiation). Since cloud fraction could only be obtained for whole days, a semi-climatological approach is used for downscaling to half-hourly values. This involved searching the time series for the 6 days with the closest mean daily $F_{ld}$ to the observed value for the given day, and taking the average of each half-hourly period for those days. While it is possible to calculate each half-hourly value using eq. 9 by assuming that cloudiness is fixed across the day, this generally produced less accurate estimates (higher RMSE). This is most likely because the empirical

parameters of eq. 10 will tend to be more accurate for daily averages because the effects of changing lapse rate and moisture profiles over the course of the day are averaged out.

### 2.2.3 Outgoing short wave radiation

Outgoing short wave radiation ($F_{su}$) is calculated from the MODIS albedo product MCD43A3 (500m 16-day) interpolated to daily frequency in combination with the incoming solar radiation data derived as previously described in section 2.1.3. For

each day, albedo is assumed invariant across all daylight periods, such that $F_{su}$ can be simply calculated from:

$$F_{su} = \alpha F_{sd}$$

(12)

Where α is short-wave albedo and $F_{sd}$ is half-hourly insolation.

### 2.2.4 Outgoing long wave radiation

Outgoing long wave radiation ($F_{ld}$) was calculated using MODIS land surface temperature product MOD11A2 (1km, daily). Observed mean daytime and nocturnal radiant surface temperatures were calculated for the site. The MODIS data were

compared to site data and linearly transformed to correct any minor biases. A semi-climatological approach similar to that described for incoming long wave radiation was used to find the 6 days with the closest daytime and nocturnal mean values to that predicted by the MODIS data. The half-hourly values for that day were calculated from the average of the 6 selected days.

### 2.3 Gap-filling of fluxes

Following the gap-filling of the meteorological drivers the fluxes of water (Fe), net ecosystem exchange of carbon (Fc), ecosystem respiration (Fre), sensible heat (Fh), and ground heat (Fg) are gap-filled using an artificial neural network (ANN)



approach. Essential background on ANNs can be found in Basheer et al. (2000) and useful examples of its application can be found in (Moffat et al. (2010) and Bryant and Shreeve (2002). We implemented the ANN as follows:

1.  gap-filling performed using the ffnet package (http://ffnet.sourceforge.net/) that is a fast and easy-to-use feed-forward neural network. The basic model we implemented is a multiple layer network with two layers of 24 and 16 nodes respectively, which creates a multilayer network that is fully connected. Training is done using a truncated Newton algorithm (TNC) to minimise a function with variables subject to bounds using gradient information. The type of ANN model and training algorithms can be altered in the code,

2.  the ANN is passed a list of inputs and outputs (which can be defined) as well as a number of iterations that are configurable in the control file (about 500-1000 is satisfactory). Additional iterations takes longer and has a tendency to over fit the model to the data. For the energy balance, the targets are processed together (Fg, Fe and Fh) and the ANN inputs are incoming solar radiation (Fsd), vapour pressure deficit (VPD), soil moisture content (Sws), Ts, Ta, and MODIS EVI (250m resolution 16 day). For Fc the inputs are Fsd, VPD, Sws, Ts, WS and EVI. DINGO will add a storage term to Fc if a profile system is present and this is a user option. Finally, the inputs for Fre are Sws, Ts, Ta and EVI. The Fre ANN is trained only using data above a u* threshold (see Section 2.3). The performance of the ANN also increased when using EVI as this provides surrogate information of vegetation activity (i.e. LAI and growth) and we hypothesise that that this equivalent to a measure of autotrophic respiration,

3.  the ANN returns a model predicted time series of data that is saved as {variable}_NN. The tower data is gap-filled using these time series and a new constructed time series is generated {variable}_Con. A QC flag is assigned as {variable}_Con_QCFlag = 1 if there is valid data from the tower, else = 99,

4.  the gap-filling frequency is configurable and can be changed to either use all data at once (ALL) or divide it into groups to process each group independently. The grouping can be all, annual, monthly OR a variable that is categorical. For example a site may be under different management at different times and the ANN can be performed separately on those periods. At some sites in the arid zone (Cleverly et al., 2013) a single ANN for the entire period does not work well so there needs to be an option to break broken down the data into monthly chunks, and

5.  the code module also outputs many diagnostic plots including general model performance (Fig. 6a), monthly time series (Fig. 6b), a 30 minute timeseries data check (Fig. 6d) and a check to see if the ANN performs diurnally (Fig. 6c).

## 2.4 Friction velocity (u*) filtering

The eddy covariance technique is well known to underestimate turbulence fluxes of carbon dioxide under stable conditions, particularly at night time where the surface can be decoupled from the measurements at a height above the canopy (Goulden et al., 1996). An excellent overview of this subject is given by Aubinet et al. (2012). This problem has been shown to impact fluxes across some Australian sites such as the old growth Mountain ash site (Kilinc et al., 2010) and a cool temperate eucalypt




forest (van Gorsel et al., 2009). The primary technique to deal with this is to exclude data taken where the eddy covariance measurements is not representative of the true flux. Typically this occurs when u* values are below a critical threshold (Goulden et al., 1996). There are several ways to calculate the friction velocity ($u^*$) threshold as shown in Aubinet et al. (2012) and in DINGO we calculate the threshold based on both the procedures of (1) Reichstein et al. (2005a) and (2) Barr et al.

(2013). Alternatively the user may select their own constant value. The threshold used for subsequent filtering is user selectable but the threshold determined using Barr et al. (2013) is used by default. Whatever choice is made the resulting u* that is used is saved to the main file named 'ustar_used'. The two methods are implemented as follows:

1. For the Reichstein et al. (2005a) approach, the non-gap-filled data set is split into 6 temperature classes of the same sample size (according to quantiles) and for each temperature class the set is split into 20 u*-classes according to

Papale et al., (2006). The threshold is defined as the u*-class where the night-time flux reaches more than 95% of the average flux at the higher u*-classes. The threshold is only accepted if the temperature class and u* are not or only weakly correlated ($|r| < 0.3$). The final threshold is defined as the median of the thresholds of the (up-to) six temperature classes. This procedure is applied to the entire dataset, giving a maximum, but conservative u* threshold (Fig. 7). The maximum value is saved as 'ustar_Reich_max'. In addition, the u* threshold is calculated continuously

using a 1 month moving window to account for seasonal variation of vegetation structure (Fig. 7) and saved as 'ustar_Reich_var'.

2. The Barr et al. (2013) approach uses a change point detection technique to objectively identify the best estimate and uncertainty range for the u* threshold. In brief, the method involves fitting a two-phase linear regression model to all possible change points within a nocturnal data sample (i.e. $2 \leq c \leq n\text{-}1$), finding the change point that minimised the

sum of squared error and establishing whether its performance was statistically significantly improved relative to a reduced form (no change point) null model. For each year, the data were divided into sequential samples of $n = 10^3$, with 50% overlap between samples. Each sample was in turn divided into four temperature classes, ordered by $u_*$ then bin-averaged ($n = 5$ for each bin) to reduce the effects of random error. To increase the sample size, the data were bootstrapped ($n = 10^3$) by simply randomly sampling (with replacement) records from the original dataset and

rerunning the analysis. This yielded a (Gaussian) distribution of u* thresholds, the mean and 95% confidence interval of which were taken as the best estimate and uncertainty of the u* threshold. As per Barr et al. (2013) we identified the dominant mode of the NEE dependency on u* (i.e. positive or negative slope below the change point), and rejected all thresholds from the non-dominant mode (in practice the negative dependency slope was very rare), and rejected any annual analysis where the number of valid change points (across all temperature strata and bootstraps) was less

than 4000 or less than 20% of the total. Annual u* statistics are saved and the annual Barr u* threshold is written to the main file as 'ustar_Barr'.



### 2.5 Calculation of Fre and GPP

#### 2.5.1 ANN modelling of Fre

Once the data have been u* filtered, they are used to train an ANN (see Section 2.2) using nighttime data only with inputs of Sws, Ts, Ta and EVI as known drivers of ecosystem respiration (Migliavacca et al., 2010). Importantly, in DINGO we also

only use flux data from the first 3 hours after sunset where the canopy is still coupled with the atmosphere, as shown in Van Gorsel et al. (2007). This makes the selection of data for the ANN model more conservative than using the entire nighttime period. This option is also user selectable. Nighttime, daytime and evening periods (first 3 hours after sunset) are determined from sunset and sunrise times calculated precisely from solar geometry accounting for elevation, latitude, longitude and day of year and saved as an integer 'day_night' (day=1, evening=2, night=3).

Initially, the Fc timeseries has nighttime gaps from the u* threshold filtering and these are filled using the ANN Fre model with the u* corrected output labelled as 'Fc_ustar'. Next, the ANN model output trained using nightime data is then extrapolated to the daytime data to create a continuous timeseries of modelled Fre (Fre_NN) that is used to fill missing gaps in the nighttime data where data where either filtered out using the u* threshold or where missing. This creates a continuous

time series of Fre (Fre_Con) that is a combination of valid observations and ANN model output. GPP is then calculated as the difference between Fc and Fre (GPP_Con) where our sign convention is –ve is a net flux into ecosystem and +ve away from it. GPP is equated to zero at nighttime but it is not forced to zero at any other time. This is because for any 30 minute value, the random error in Fc results in values of Fc being higher or lower than the true value. Due to this error, it is possible then to get values of GPP that have either a higher or lower random error component. There is also no random error in the Re ANN

calculation and if we force these GPP values to zero then there is a bias in the GPP integrals (e.g. daily totals) because we only remove the values that have a negative error. Forcing GPP to zero during the daytime leads to ever so slightly higher GPP because lower values are cut off particularly in the evening and morning when GPP approaches zero.

#### 2.5.2 Separation of Fc using a light response curve approach

An additional approach is also applied where a simple parametric approach to the imputation of Fc is undertaken in DINGO, and largely follows the approach of Lasslop et al. (2010). In this approach, Fc is modelled as the sum of gross primary production (GPP) and ecosystem respiration (Fre):

$$Fc\ =\ GPP\ +\ Fre \tag{13}$$

A Michaelis Menten-type simple rectangular hyperbolic light response model (Ruimy et al., 1995) of modified form (as per Falge et al. (2001)) is used in conjunction with the Arrhenius-style temperature response function as proposed by Lloyd and

Taylor (1994), such that GPP and Fre are replaced by:



$$Fc = \frac{\alpha Q}{1 - Q/2000 + \alpha Q/\beta} + rb\, e^{E_o\left(\frac{1}{T_{ref} - T_0} - \frac{1}{T - T_0}\right)} \qquad (14)$$

Where α is the initial slope of the photosynthetic light response, Q is photosynthetic photon flux density, and β is photosynthetic capacity at 2000 μmol photons m$^{-2}$ s$^{-1}$, *rb* is the reference respiration at a reference temperature (T$_{ref}$ – here set to 10ºC), E$_o$ is an activation energy parameter that determines the function's temperature sensitivity, and T$_0$ is the temperature at which metabolic activity approaches zero.

Lasslop et al. (2010) proposed an additional criterion – adopted here - to take account of the effect of vapour pressure deficit (VPD) on stomatal conductance - and thereby photosynthetic capacity – in which there is a non-linear decline in photosynthetic capacity once VPD exceeds a given threshold, as follows:

$$\beta = \begin{cases} \beta_0\, e^{(-k(VPD - VPD_0))}, VPD > VPD_0 \\ \beta_0, VPD < VPD_0 \end{cases} \qquad (15)$$

As has been widely reported, even in isolation the unconstrained temperature response function is over-parameterised (Lloyd
and Taylor, 1994; Reichstein et al., 2005b; Richardson and Hollinger, 2005). Thus T$_0$ and T$_{ref}$ are fixed at -46.02ºC and 10ºC, respectively, as per Lloyd and Taylor (1994). E$_o$ is fitted nocturnally using the data for each year, as per Reichstein et al. (2005a). The default fitting window for the remaining parameters is 15 days and the default time step is 5 days (missing dates are then interpolated to generate a complete time series of daily parameter estimates), but both of these can be configured (minimum of 1 day, maximum of 30 days in each case). The quality control scheme for parameter acceptance used by Lasslop
et al. (2010) was adopted for DINGO.

The user can make two choices: i) optimise all remaining parameters (α, β, k, rb) using the daytime data alone, or; ii) optimise rb using nocturnal data and optimise light response parameters (α, β, k) using daytime data. The first choice may be selected where difficult nocturnal conditions limit the amount of data available for robust respiratory parameter estimation. However,
simultaneous optimisation of all parameters may result in unrealistic estimates when signal magnitude is weak. Random error in eddy covariance data is heteroschedastic (i.e. random error magnitude increases with flux magnitude), and is non-zero at zero flux (Richardson and Hollinger, 2005). Thus signal: noise is likely to be lower for low productivity ecosystems, which are common in Australia due to soil moisture and nutrient limitation. As such, the second choice may be more appropriate for many sites.

Once complete daily time series for the parameter estimates, eq. 14 is used in conjunction with gap-filled driver data to calculate half-hourly values for Fc, GPP and Fre that are named as Fc_Lasslop, GPP_Lasslop and Fre_Lasslop, respectively. The drivers used are also user configurable – for example, either incident or absorbed photosynthetically active radiation can be used for GPP parameter estimation and calculation, and soil or air temperature (or a weighted combination thereof) can be
used for Fre.



### 2.6 Uncertanities

In version 12a, the only estimates of uncertainty are a simple sensitivity to changing the chosen u* threshold above (130%) and below (80%, 60%, 40%, 20% and no threshold) the determined threshold. The carbon fluxes (GPP, Fc and Fre) are

recalculated as above and annual sums given for each of the threshold manipulations and saved as a separate output. In version 13, DINGO will calculate a subset of annual Fc uncertainties associated with: 1) $u_*$ threshold estimation error; 2) random error, and; 3) imputation error and these are described below:

1. the most straightforward approach to propagating $u_*$ threshold uncertainty to annual Fc is simply to run the gap-filling procedure for Fc after filtering the nocturnal data using the upper and lower bounds of the 95% confidence interval

for the threshold derived from change point detection (see section 2.4). However, because a much larger proportion of nocturnal data is removed when filtering for the upper bound than for the lower bound, the proportions of observational and gap-filled data necessarily change, and the random and model error uncertainty contributions thus vary depending on the $u_*$ threshold. For this reason, a full Monte Carlo-style simulation (described below) is required to simultaneously account for all of the above error sources.

2. we adopted the daily differencing approach of Hollinger and Richardson (2005). This method assumes that when differences in critical drivers are sufficiently small (<35 Wm$^{-2}$ for insolation, <3 ℃ for air temperature, and <1 m s$^{-1}$ for wind speed), differences in estimates of Fc separated by 24 hours represent random error. Since random error is heteroschedastic, its magnitude (expressed by the standard deviation or σ[δ], since the mean is expected to be zero)

must be calculated for ranges of $u_*$, and expressed as a function of the mean flux magnitude over that range. Thus a linear relationship between σ(δ) and flux magnitude is generated. Since the random error distribution for eddy covariance data is Laplacian rather than Gaussian, a random error estimate for a given datum can be generated from a Laplacian random error distribution with location parameter of zero and scaling parameter of σ(δ) / √2. To compound the uncertainty on each point to annual Fc, DINGO runs a Monte Carlo-style simulation: for each of $10^4$

trials (for each year in the dataset), a realisation of random error is generated for the entire observational Fc time series, and the annual sum of the observational data is calculated. The 95% confidence interval for the resulting distribution of Fc sums is returned as the uncertainty due to random error.

3. DINGO adopts the approach of Keith et al. (2009), separately for day and night conditions, a sub-sample of $10^3$ observations is randomly selected from the annual dataset. Gaps are then introduced into this dataset such that the

proportion of missing data is equal to the observed proportion of missing data annually. The missing data is filled with model estimates, and the percentage difference between the complete observational and gap-filled time series' is calculated. As with random error, this procedure is run $10^4$ times. The 95% confidence interval for the distribution of the percentage difference estimates is calculated, and these percentages are applied to the annual sum (for example,



if the 95% CI is -5% to +5%, then a carbon sink of 500 gC m⁻² a⁻¹ has a model-induced uncertainty of ± 25 gC m⁻² a⁻¹).

To combine uncertainties for random and model error alone, we assume their independence and sum in quadrature:

$$\varepsilon_{tot} = \sqrt{\varepsilon_r{}^2 + \varepsilon_m{}^2} \qquad (16)$$

Where $\varepsilon_{tot}$, $\varepsilon_r$ and $\varepsilon_m$ are combined total, random and model uncertainty, respectively. It must be emphasised that this is not strictly valid, since the model error uncertainty quantification above necessarily includes some effects of random error (since the observational input data includes random error). While it is possible to separate model and random error components (see Dragoni et al. (2007), for example), this requires an accurate estimate of random error. It has been noted elsewhere (Billesbach, 2011; Dragoni et al., 2007; Hollinger and Richardson, 2005) that the above method for deriving random error tends to

overestimate, since some signal is likely to be included with noise during the differencing procedure, and wind direction is neglected (whereas the source-sink may be azimuthally variable). We argue that since this is likely to result in overestimation of error, the approach is conservative.

To combine all three sources of uncertainty, DINGO runs $10^4$ trials, in each of which the following steps occur:

1. an estimate of u* threshold is randomly drawn from the previously derived u* distribution,

    2. data are screened for nocturnal conditions below this threshold,

    3. Fc model estimates are generated and used to fill the missing data,

    4. the random error uncertainty contribution is calculated using a single realisation of random error,

    5. the model error uncertainty is calculated using a single realisation of model error,

6. 3, 4 and 5 are summed, and the resulting annual Fc is recorded.

The total uncertainty is then calculated as the 95% confidence interval for all annual Fc estimates. Since model optimisation and gap filling must be run for the entire dataset for each of these trials, this approach is computationally expensive and correspondingly time consuming. As such, it is possible to switch each of the uncertainty algorithms on or off.

**2.7 Flux footprint calculation**

In version 13, DINGO allows for derivation of flux footprints to assist the user in interpretation of flux results. The two-dimensional footprint parameterisation of Kljun et al. (2015) is implemented in DINGO, providing the possibility to calculate footprint estimates for convective to stable conditions and any measurement height within the planetary boundary layer. As any of the currently available fast footprint models, the applied footprint parameterisation assumes stationarity and horizontal

homogeneity of the flow within the 30-min averaging time period for flux calculations. Hence, the footprints provide only an



approximate guidance for flux towers within strongly heterogeneous terrain. The user can select whether or not flux footprint estimates should be produced. The footprint model is implemented as follows:

1. footprint model input values are supplied as gap-filled meteorological drivers,

2. the extent of the 80% footprint and the peak location of the crosswind-integrated footprint are calculated for each 30-min time step. Two-dimensional footprints for each 30-min time step are rotated into the mean wind direction of the according time step,

3. two-dimensional footprint function values are aggregated over a time interval selected by the user, e.g. monthly or annual intervals, to produce so-called footprint climatologies,

4. footprint results are provided in raster format that can be merged with surface imagery or can also be used to calculate footprint-weighted surface characteristics (e.g. footprint-weighted contribution from land cover classes surrounding the flux tower). The raster resolution and extent can be set by the user, and

5. by default, DINGO provides the extent of the 80% footprint and the peak location of the crosswind-integrated footprint, as well as plots of annual footprint climatologies.

## 2.8 Diagnostics and results

DINGO produces a variety of diagnostic, summary and results plots that assist the user in immediate visualisation of the data. These plots enable rapid identification and correction of instrument or processing errors. The suite of outputs includes:

1. plots for the identification of energy balance non-closure (after Franssen et al., 2010; Leuning et al., 2012; Oken, 2008) showing scatter plots of the difference between turbulent fluxes and available energy for; a) all hours of 30 minute values, b) daytime hours of 30 minute values, c) nighttime hours of 30 minute values and d) daily means. An example is shown in Fig. 8.

2. plots that show the difference between actual tower observations and best alternate time series for each variable such as those constructed using BoM AWS station data or the ANN model output. This essentially compares what we think the variable 'should' be with what it really is. If there is a big difference then it could indicate instrument or processing differences that may need require further examination.

3. calculation and plots of weekly timeseries of ecosystem scale water use efficiency (WUE) following Beer et al. (2009), radiation use efficiency (RUE) following Garbulsky et al. (2010), energy balance closure following (Twine and Kucharik, 2008), evaporative fraction (EF) following (Zhou and Wang, 2016) and the Bowen ratio (BR) (Bowen, 1926). These plots can be used to identify physically and physiologically inconsistent data periods.

4. graphs indicating the missing data for all variables including percentage of each month that is gap-filled and the percentage of any data not gap-filled (Fig. 9a) and monthly time series plots where data with more than 30% of data gap-filled is shown in light blue (Fig. 9b).





5.  summary figures of variables in fingerprint style for non-gap-filled and gap-filled variables using code from OzFluxQC (Isaac et al., 2016) (Fig 10).

6.  summary timeseries plots of daily means and a 30 day running mean of net ecosystem exchange (Fc), ecosystem respiration (Fre) and gross primary production (GPP) (Fig. 11).

7.  results showing the cumulative carbon (GPP, Fc and Fre) and water (Fe and precipitation) by year (Fig. 12).

In addition, DINGO allows for the output of data in text based CSV format for a variety of purposes that is user selectable. Different outputs include files that are ready to be used in the online eddy covariance gap-filling & flux-partitioning tool (EddyProc) (http://www.bgc-jena.mpg.de/~MDIwork/eddyproc/) or processing to daily files for use by the National Aeronautics and Space Administration for satellite validation projects such as Spaceborne sun-induced vegetation fluorescence validation (Sanders et al., 2016).

**3 Conclusions**

The OzFlux network has been highly successful in generating standardised measurements and protocols that provide robust primary data. Only via transparent, advanced and consistent QA/QC will we ensure compatibility within the OzFlux network (Beringer et al. 2016) and with international databases (FLUXNET) (Papale et al., 2006), ensuring uptake by the broader scientific community. Through robust software systems such as OzFluxQC (Isaac et al., 2016) and DINGO we are able to ensure timely and quality gap-filling and partitioning of fluxes that in turn enables a significant uptake of the eddy covariance data for application to a range of research questions as exemplified in Beringer et al. (2016). This includes integration of eddy covariance and remote sensing datasets for the validation of satellite products (e.g. GPP and ET (Kanniah et al., 2009; Restrepo-Coupe et al., 2015)) and to aid the parameterisation of models that rely on remotely sensed data (e.g. GPP, ET, canopy conductance, and light use efficiency (LUE) (Barraza et al., 2014, 2015; Glenn et al., 2011; Goerner et al., 2011)). In addition, OzFlux data have been instrumental in constraining a continent-wide assessment of terrestrial carbon and water cycles (Haverd et al., 2013a) and featured in the development of new models (Haverd et al., 2007, 2009). There is utility of the data to support carbon accounting type activities (Hutley et al. 2005), as demonstrated in research focussed on the conversion of savanna to pasture (Bristow et al. 2016). However, as for many applications like this, a measure of uncertainty of the fluxes are required (Schmidt et al., 2012) and we see this as a major opportunity for further development of DINGO. In addition, the integration of flux data must have due consideration for scale and the associate footprint of the flux measurements for correct interpretation. We also see this as a feature that would be useful for users and therefore we also plan to incorporate into V13. Ultimately flux data are required to address the key ecosystem science questions of OzFlux (Beringer et al., 2016) that are focused on improved understanding of the responses of carbon and water cycles of Australian ecosystems to current climate and disturbance regimes as well as impacts of projected future changes to precipitation, temperature and atmospheric $CO_2$ concentration. Key questions include 1) what are the key drivers of ecosystem productivity (carbon sinks) and greenhouse gas





emissions; 2) how resilient is Australian ecosystem productivity to an increasingly variable and changing climate, and 3) what is the current water budget of the dominant Australian ecosystems and how will it change in the future?

**Acknowledgments**

5    This work utilised data collected by grants funded by the Australian Research Council DP130101566. Beringer is funded under an ARC Future Fellowship (FT110100602).



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





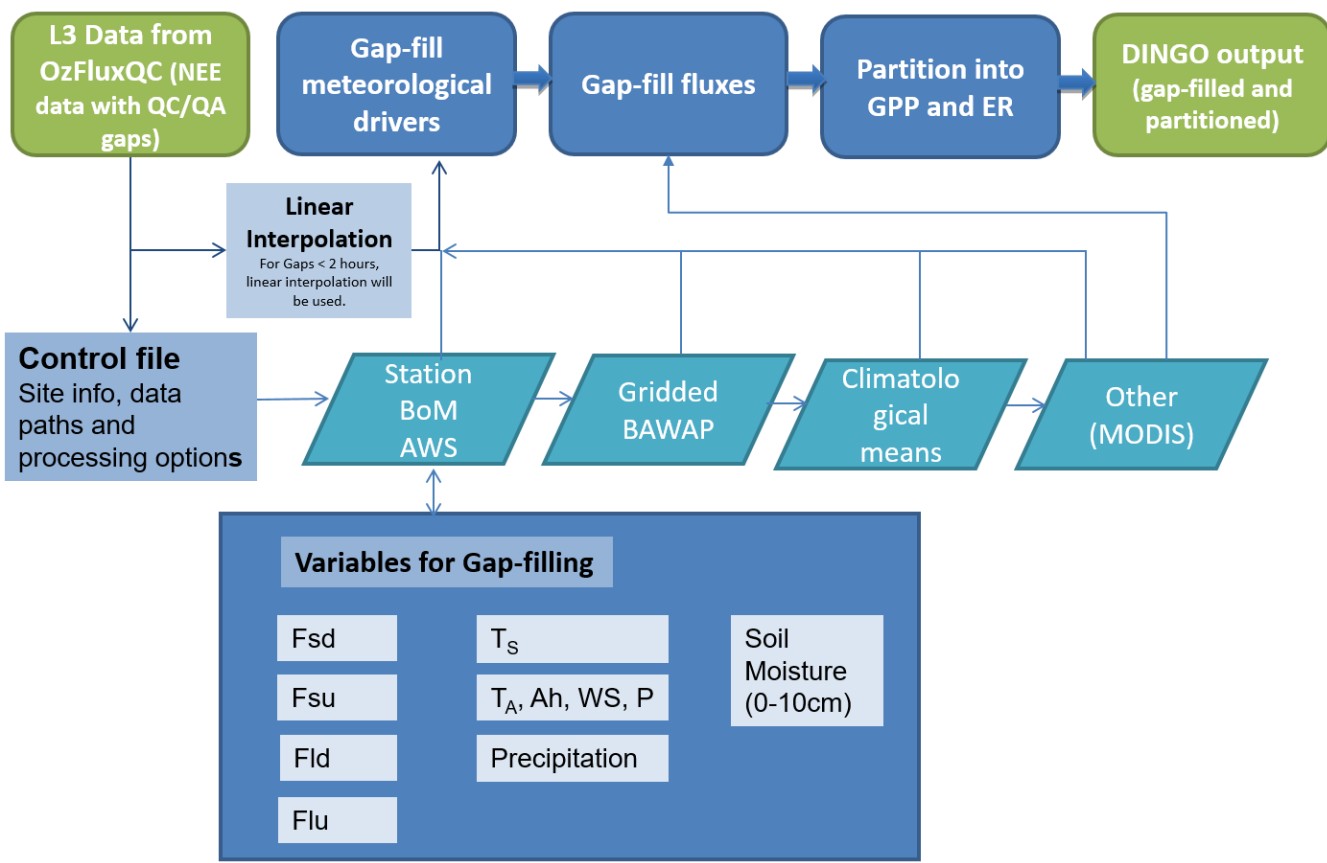

**Figure 1: Overview of processing pathways in Dynamic INtegrated Gap-filling and partitioning for Ozflux (DINGO) from level 3 OzFluxQC data to gap-filled and partitioned outputs. Where BoM is the Australian Bureau of Meteorology, AWS is Automatic Weather station, BAWAP is gridded meteorological data at 0.1 degree resolution (Bureau-of-Meteorology, 2013), MODIS is the Moderate-resolution imaging spectroradiometer on-board the Terra and Aqua satellites, Fsd is incoming shortwave radiation, Fsu is reflected shortwave radiation, Fld is incoming longwave radiation, Flu is emitted longwave radiation, Ts is soil temperature, Ta is air temperature, Ah is absolute humidity, WS is wind speed, P is atmospheric pressure, GPP is gross primary productivity, ER is ecosystem respiration.**





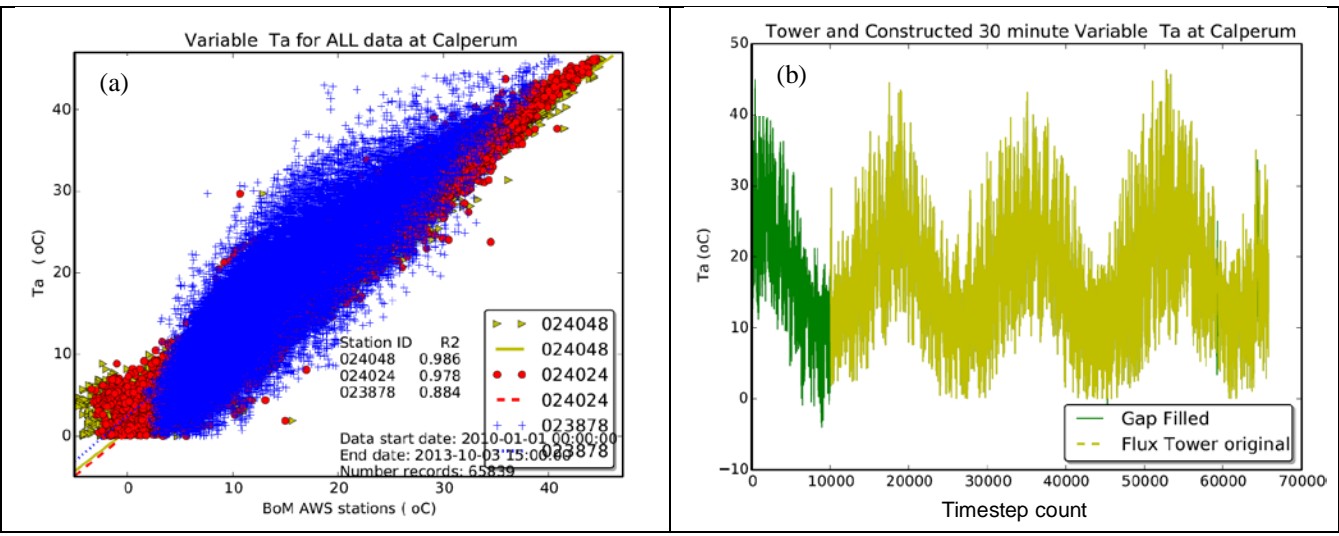

**Figure 2: An example of gap-filling meteorological variable (air temperature (Ta – °C) in this case) using nearby Bureau of Meteorology automatic weather stations (AWS). Example is from the Calperum flux tower (see Beringer et al. (2016) for details) for 2010 to 2013. Figure illustrates a) the correlations of site Ta with the best nearby AWS including station ID and $r^2$ and b) the final gap-filled Ta series of 30 minute data and the original flux data with gaps.**



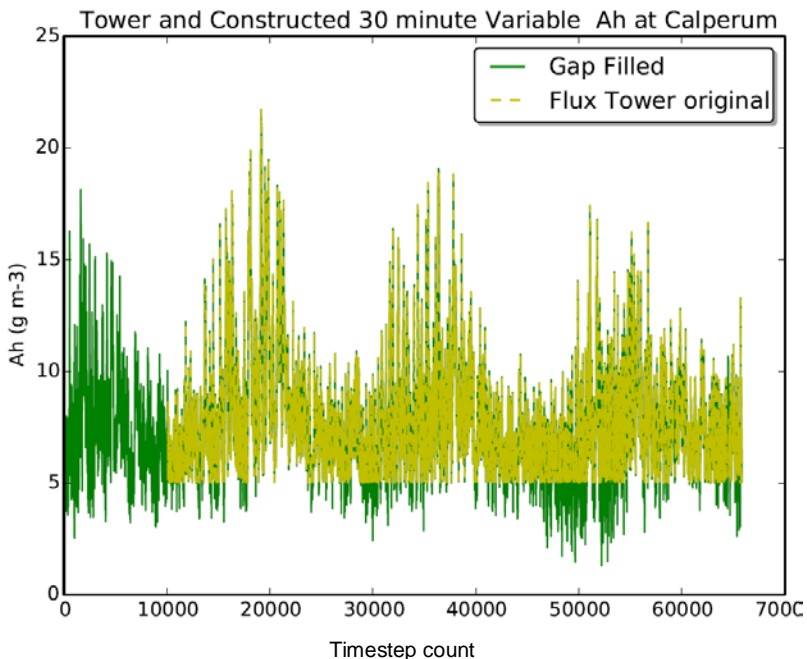

**Figure 3: Diagnostic plot of absolute humidity (Ah - g m⁻³) showing the time series used for gap-filling (green) and noticeably the flux tower data from OzFluxQC level 3 that has data cut off due to incorrect setting of threshold for quality control. Following this the user would go back to the OzFluxQC processing and correct the error. Example is from the Calperum flux tower (see Beringer et al. (2016) for details) for 2010 to 2013.**



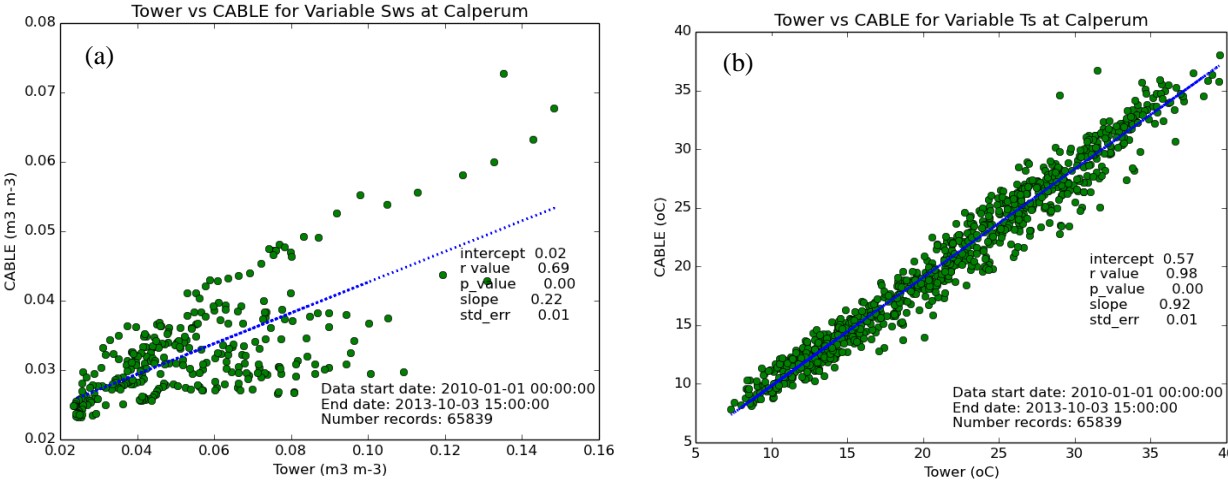

**Figure 4: An example of gap-filling of a) soil moisture (Sws – m³ m⁻³) and b) soil temperature (Ts – °C) using output from the BIOS2**
**model as described in (Haverd et al., 2013a, 2013b) forced using remotely sensed estimates of leaf area index (LAI) and meteorology**
**from the Bureau of Meteorology's Australian Water Availability Project data set (BoM AWAP) (Jones et al., 2009). A regression**
**equation is calculated from the modelled data versus the available site data and then applied to adjust the model time series. The**
**modelled time series is then used for gap-filling and applied identically as for the meteorological drivers to produce gap-filled soil**
**variables. Example is from the Calperum flux tower (see Beringer et al. (2016) for details) for 2010 to 2013.**





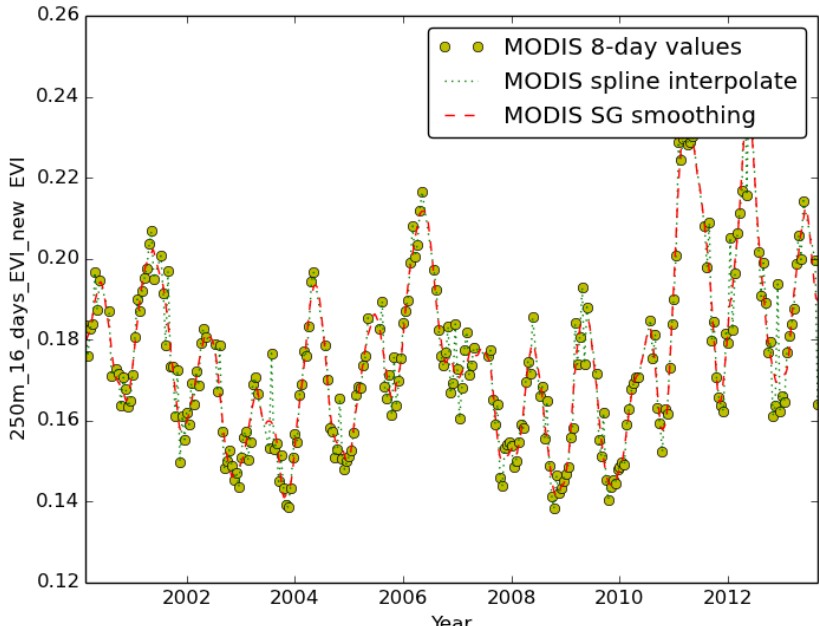

**Figure 5: An example of ingestion and processing of MODIS satellite information. Here MOD13Q1 Enhanced Vegetation Index (EVI) for the 3x3 km cut-out around the tower is illustrated. The Modis eight day values are shown as circles and data is then interpolated (green line) and smoothed (red line) as detailed in Section 2.1.4. Example is from the Calperum flux tower (see Beringer**
5  **et al. (2016) for details) for 2010 to 2013.**



**Figure 6: Gap-filling of fluxes is under taken as in Section 2.2. DINGO produces a number of diagnostic plots for each target. In this case net ecosystem exchange (Fc - µmol m$^{-2}$ s$^{-1}$) is shown. The plots allow the user to assess the ANN model performance a) against flux tower data, b) at seasonal (monthly) timescales, c) at diurnal timescales and d) 30 minute timescales. Example is from the Calperum flux tower (see Beringer et al. (2016) for details) for 2010 to 2013.**





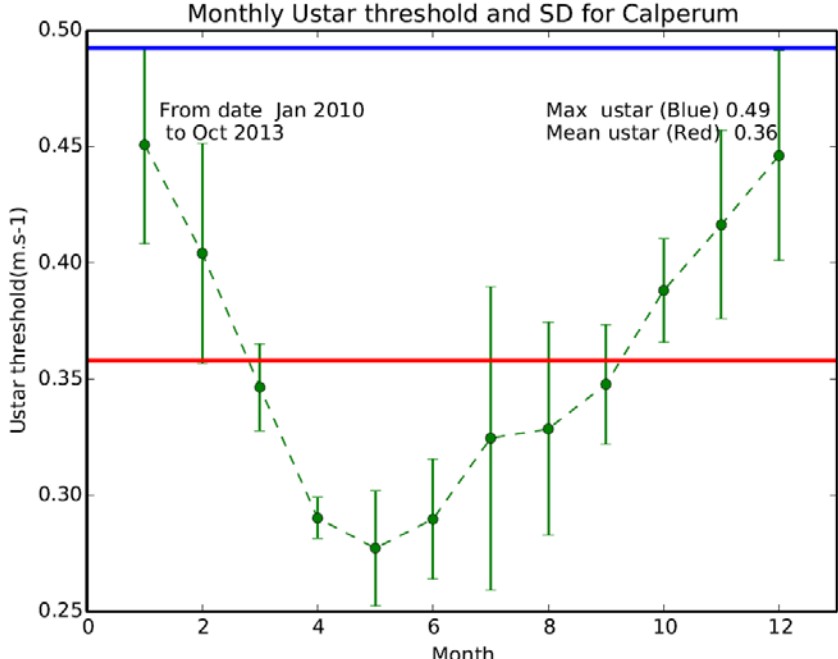

**Figure 7: The u\* threshold is determined as given in Section 2.3. Shown here is u\* determined using the procedure of Reichstein et al. (2005a). The u\* is shown for the 3 month moving windows (green line with standard error bars indicating annual variability). Also illustrated are the maximum u\* thresholds over the entire period (conservative, blue line) and mean u\* over all monthly bins (red line). Example is from the Calperum flux tower (see Beringer et al. (2016) for details) for 2010 to 2013.**







**Figure 8: DINGO produces many diagnostic plots including energy balance closure (difference between Fh+fe and Fn-Fg) for a) all hours of 30 minute values  b) daytime hours of 30 minute values c) nightime hours of 30 minute values d) daily means. Fh is sensible heat flux (W m$^{-2}$), Fe is latent heat flux (W m$^{-2}$), Fn is net radiation (W m$^{-2}$) and Fg is ground heat flux (W m$^{-2}$). Example is from the Calperum flux tower (see Beringer et al. (2016) for details) for 2012.**





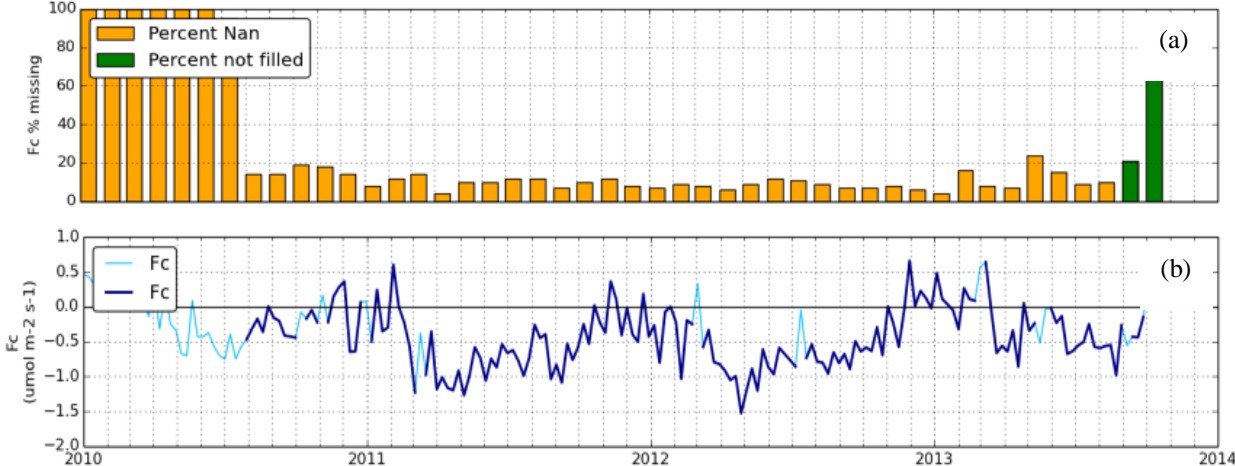

**Figure 9: DINGO produces many diagnostic plots for all fluxes and meteorological variables (in this case net ecosystem exchange**
**(Fc – μmol m$^{-2}$ s$^{-1}$)) including a) amount of missing data and data not gap-filled and b) weekly plot with period of more than 30%**
**missing data shown in light blue. Example is from the Calperum flux tower (see Beringer et al. (2016) for details) for 2010 to 2013.**





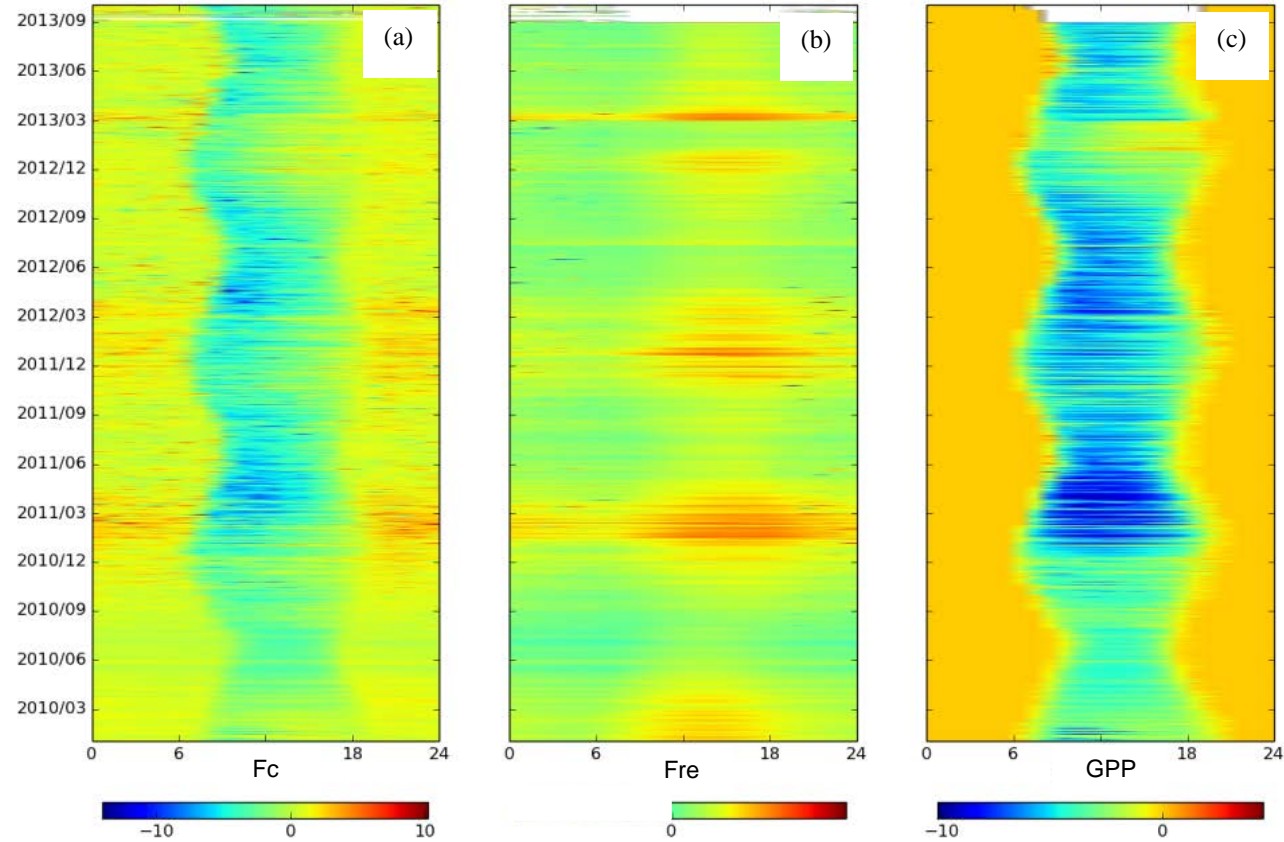

**Figure 10: Example of summary fingerprint plot of a) net ecosystem exchange (Fc), b) ecosystem respiration (Fre) and c) gross primary production (GPP) all in units of µmol m⁻² s⁻¹. Example is from the Calperum flux tower (see Beringer et al. (2016) for details) for 2010 to 2013.**



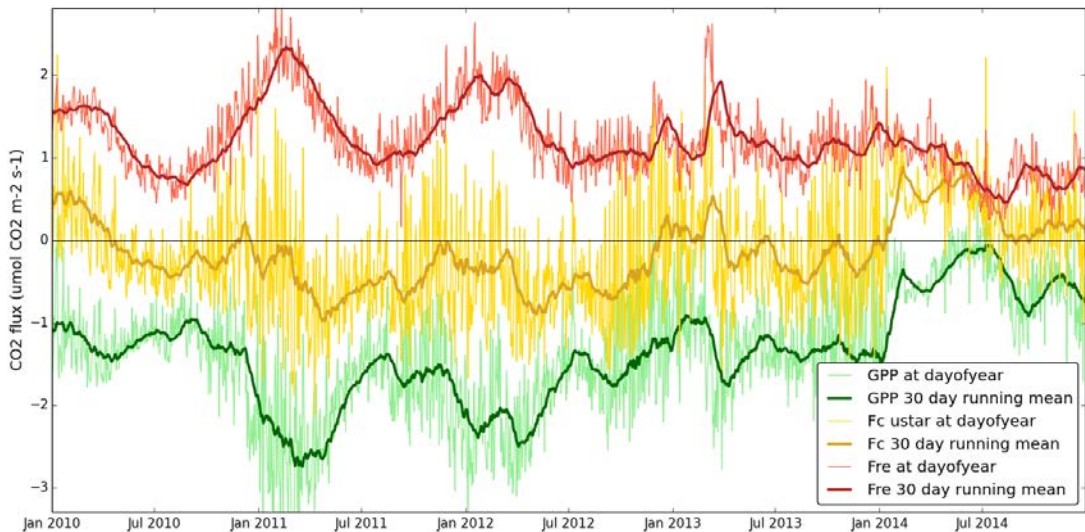

**Figure 11: Summary timeseries plot of daily means and a 30 day running mean (dark line) of net ecosystem exchange (Fc, yellow), ecosystem respiration (Fre, red) and c) gross primary production (GPP, green) all in units of µmol m$^{-2}$ s$^{-1}$. Example is from the Calperum flux tower (see Beringer et al. (2016) for details) for 2010 to 2014.**





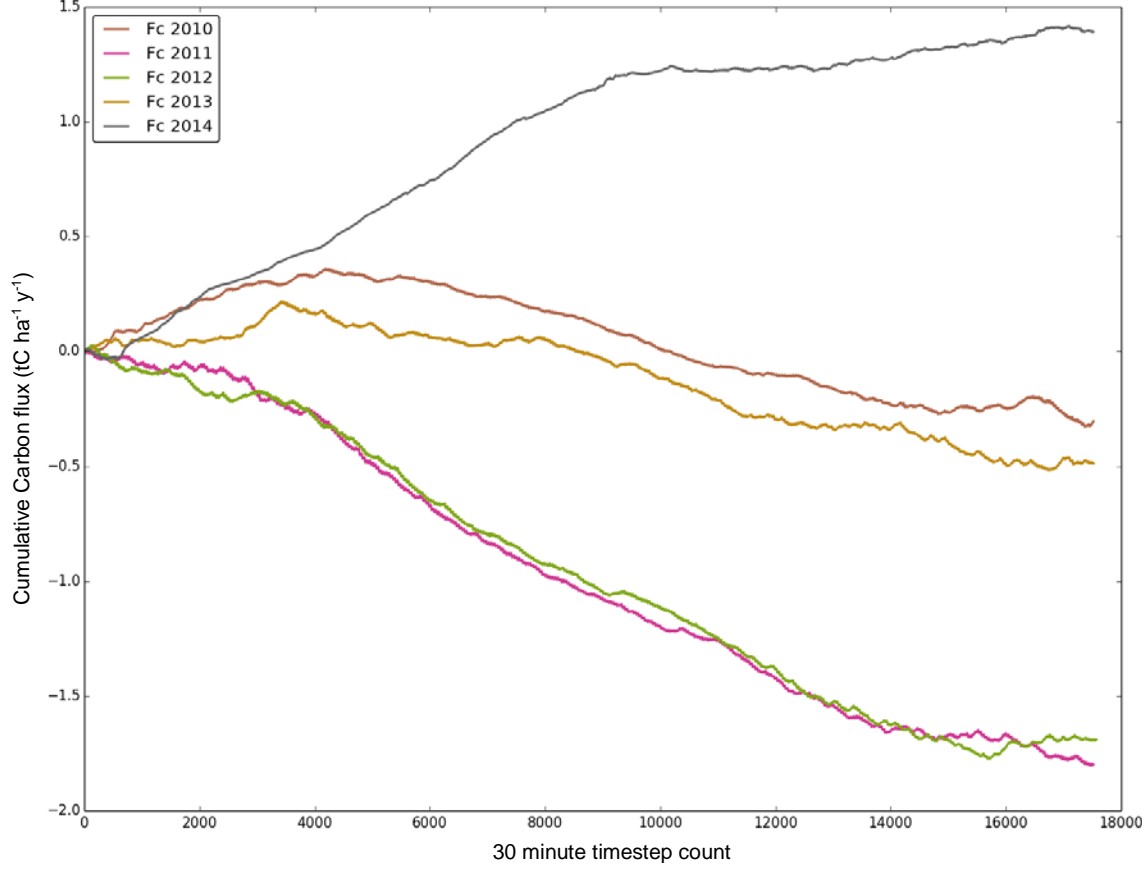

**Figure 12: Example of results plot of cumulative carbon uptake (t C ha⁻¹ yr⁻¹) (negative being uptake) by year. Example is from the Calperum flux tower (see Beringer et al. (2016) for details) for 2010 to 2014 that in this case shows the system swings from being a sink to source from one year to the next.**

