# Peer review of "Technical note: Dynamic INtegrated Gap-filling and partitioning for OzFlux (DINGO)"

_Biogeosciences, 2016_

## Referee Comment (RC1) · Anonymous Referee #1 · 16 Jun 2016

The manuscript is well written and describes a useful system to automate gapfilling. I do not find anything radically new in the approach. Each step in the process is based on well known, and widely used approaches to gapfilling. The novelty, perhaps, is in the automation of the process that offers new ease of use and the potential for widespread application. The test for such a new automated process is the applicability and its wide usability. Many flux sites have such a process for their own site, and none (or few) considered publishing it. As long as the process is specific to only and nothing more than OzFlux, I do not think it makes an interesting scientific paper, and should be published as an OZFlux report or the DINGO software manual. This could be easily remedied, and I propose that the authors will make the effort to generalize DINGO so that it could be applied by any flux site that desires to.

You write in the abstract that "DINGO was developed for Australian data but the framework is applicable to any flux data or regional network". However, it seems to be hard-coded for the Australian data sources and therefore will technically no run for other sites. As there is no novelty to the formulation, I see one of the big advantages in the automation of the work process and particularly in the data retrieval for the gap-filling, which is usually a cumbersome and laborious process. Currently, when gap-filling the forcing data the software only reads Australian met data and Australian high resolution meteorological reanalysis. Is it possible to allow a generic, user provided, table with data from n met stations nearby? Each user can prepare the data from their local met stations but the process of data selection and correlation with the flux-site met obs can be done by DINGO. Can DINGO be made to access the ECMWF reanalysis? It is coarser than the Australian one, but global and would allow users worldwide to have an alternative to local met stations.

For precipitation, which is very poorly generalized from one met station to the other (and particularly the time of precipitation) I suggest that even for the OzFlux data processing there could be a great advantage in using satellite-derived precipitation from TRMM or the new GPM products.

Similarly, the soil moisture model can offer the option to be driven by MODIS LAI and ECMWF as an alternative to the strictly Australian data sources. That should be easy as MODIS is extracted anyhow for the albedo gapfilling.

There is nothing new in incorporating footprint and estimating the uncertainty of the gap-filling method. In fact, it is irresponsible not to do so. These are features that you say will be available in V13, but I see no purpose in publishing an application to a well-known method that is not done yet. However, you say that version 13 is due July 2016. This is a few weeks from now, and before the revision will be due. Can you make this paper about the complete and fully functional version 13?

Other comments: Why are you using different meteorology to drive the soil water model (meteorology from the Bureau of Meteorology's Australian Water Availability Project)

[Figure]

than the station and other met data already processed and gapfilled for the station location, if station data is missing?

The u* filtering section should come before the ANN gap-filling section as u* filtering creates most of the gaps that then need to be filled.

---

## Referee Comment (RC2) · Anonymous Referee #2 · 17 Jun 2016

GENERAL COMMENTS

This manuscript describes a dynamic integrated gap filling and partitioning tool (DINGO) developed for standardized processing of OzFlux station data. The tool is designed to gap fill both meteorological variables and fluxes, besides it partitions the net flux into its components (Gross Primary Productivity and Ecosystem Respiration). Although it raises the important issue of a standardized post processing for meteorological and eddy flux data at network level, unfortunately the manuscript doesn't introduce important novelty aspects, and it sounds more like a technical note than a scientific contribution. It provides a very detailed description of the tool, but lacks of analysis and interpretation. Besides, most of the figures included in the manuscript are standard diagnostic or result plots provided by the tool itself and not the result of an in depth analysis of the tool behavior, performance or applicability. Some important aspects that

should be faced and deepened are not included, as for example: the evaluation of the tool performances and their dependence on gap percentage, length and distribution, a comparison with other existing gap-filling and partitioning tools, an analysis of the effects of the gap filling on annual sums... Results of the uncertainty estimation would also enrich the work, together with some performance analysis at site/PFT level. A technical observation: u* filtering should be applied before gap filling since it introduces gaps in the time series. For the above mentioned reasons, I do not think that the manuscript, in its present form, is suitable for publication in Biogeosciences.

---

## Referee Comment (RC3) · Anonymous Referee #3 · 17 Jun 2016

The manuscript presents an overview of the software framework and methods for automated gap-filling and partitioning of OzFlux tower network data. Overall, the manuscript does a nice job presenting this overview with the right amount of detail and rationale, and is well-written. The gap-filling and partitioning methods employed are generally state-of-the-art community standards. That said, there is not much new information here. Perhaps the greatest novelty lies in the gap-filling of ancillary drivers, where to my knowledge the only other recent work on this topic is that of Vuichard and Papale (2015). Although that work is not referenced, the present manuscript expands upon it to incorporate data from nearby weather stations and satellite remote sensing observations, with additional soil- and radiation-tailored gap-filling methods.

I have two major recommendations: 1) The manuscript could be made more impactful by including an analysis of performance of the gap-filling and partitioning results

across all or part of the OzFlux network. This would demonstrate the utility/flexibility of the framework, its ability to address the grand science challenges identified in the introduction, and highlight areas for continued improvement. 2) The paper would benefit from a discussion of how the methods employed in this paper conform to or push the envelope of current community practice. This would clearly communicate the significance of the software suite and its novel contributions.

Specific comments:

- The representation of meteorological quantities should make more effort to be consistent with community standards. I have found Reifsnyder et al. (1991) particularly useful for this purpose, presenting the symbols, units, and notation for use in the journal Agricultural and Forest Meteorology. Conforming to these where possible/practical will improve the readability and reach of the manuscript. Also, both Fre (pg. 3, line 31) and ER (Fig. 1) are used in the manuscript to represent ecosystem respiration, and both Fc (pg. 6, line 14) and NEE (Fig. 1) are used to represent net ecosystem exchange of CO2.

- Pg. 5, line 17-20 & Pg. 6, line 9-11: Is there a threshold correlation below which the data are not gap-filled? A discussion of this topic is warranted.

- Pg. 5, line 17-20: Linear regression is a good start for using nearby station data for gap-filling. However, even from Fig. 2a (which looks to be a very clean example), the best correlation has a distinct non-linear component at the low end of values. Using this relationship to gap-fill the time series then extends the data into minimums not actually observed at the site, which has the potential to influence down-stream modeled physiological responses. Perhaps the manuscript could address this topic as a discussion point, and/or include a few different fitting functions in future iterations (maybe piecewise linear fits?).

- Pg. 8, line 9-12: It would be informative to show some stats addressing how well this procedure replicates actual variation in solar radiation under cloudy conditions across

the OzFlux network (opposed to using nearby BoM site data). What about using the diurnal average approach employed for estimating incoming longwave radiation under cloudy conditions (pg. 9, line 6-8)?

- Pg. 10, line 4-5: How was the ANN hidden layer architecture of 24 and 16 nodes arrived at? Appropriate model complexity can significantly impact ANN model performance. Recent related works using ANNs for gap-filling have tested for ideal architectures (Papale and Valentini, 2003; Knox et al. 2015; Baldocchi et al., 2016).

- Pg. 10. ANN procedure: How are the data split into training and testing/validation sets?

- Pg. 12, line 12: "u* corrected output" implies that that data were somehow corrected for low u* conditions. I recommend rephrasing.

- Pg. 12, line 16: Please clarify which Fre variable was used (Fre_NN or Fre_Con).

- Pg. 12, line 16: What does "ve" mean? (as in +ve, -ve)

- Pg. 12, line 17-22: This explanation is confusing. Recommend rephrasing. Consider explaining that forcing GPP to zero at night removes positive and negative random error equally, but forcing any positive GPP values to zero during daytime would bias results because only negative random errors would remain. Also, I am not sure what is meant by "no random error in the Re ANN calculation", since random error is relevant in any model fit to real data.

- Pg. 14, line 16-27: Since the ANN is employed for gap-filling, why not use the model residuals as an estimate of the random error? The model residuals of high performance gap-filling algorithms such as the ANN provide a good, if not conservative estimate of the random uncertainty (Moffat et al. 2007, Richardson et al. 2008). The daily differencing approach is much more conservative as it includes natural environmental variability as a result of variation in the flux footprint. This would help alleviate some of the double-counting of uncertainty mentioned in a later paragraph (pg. 15, line 7-12).

- Pg. 14, line 19: Please indicate what lowercase delta means.

- Pg. 14, line 18-21: Why must random error be calculated over bins of u*? (rather than over bins of (or a regression with) flux magnitude)

- Section 2.6 Uncertainties: This section would benefit from a discussion on the uncertainty propagated along the entire processing chain described in the manuscript (i.e. using gap-filled ancillary drivers value to gap-fill fluxes).

- Pg. 17, line 16-18: What is the suggested turnaround time from data collection to quality output from the procedure outlined in the manuscript?

Technical corrections

- All bulleted lists: In some cases the lack of sentence case following the bullet point makes sense. However, in many cases the sentence following the bullet is a stand-alone sentence and should use sentence case.

- Pg. 10, line 29: "stable conditions" should specifically reference "atmospherically stable conditions"

- Pg. 13, line 26: I am confused by the sentence fragment "Once complete daily time series for the parameter estimates,...". What about this: "Once daily estimates for the parameters are generated,..."

- Labels in figures need cleaning up (e.g. Fig. 4 y-axis – what is CABLE?, Fig. 5 y-axis – make this human readable, Fig. 9b legend – what is the difference between Fc and Fc?, use superscript and subscripts where appropriate)

- The manuscript needs thorough editing for typographical errors

References cited

Baldocchi, D. et al. The impact of expanding flooded land area on the annual evaporation of rice. Agricultural and Forest Meteorology 223, 181–193 (2016).

Knox, S. H. et al. Agricultural peatland restoration: effects of land-use change on greenhouse gas (CO2 and CH4) fluxes in the Sacramento-San Joaquin Delta. Global Change Biology 21, 750–765 (2015).

Moffat, A. M. et al. Comprehensive comparison of gap-filling techniques for eddy covariance net carbon fluxes. Agric. For. Meteorol. 147, 209–232 (2007).

Papale, D. & Valentini, A. A new assessment of European forests carbon exchanges by eddy fluxes and artificial neural network spatialization. Global Change Biology 9, 525–535 (2003).

Reifsnyder, W. E., McNaughton, K. G. & Milford, J. R. Symbols, units, notation. A statement of journal policy. Agricultural and Forest Meteorology 54, 389–397 (1991).

Richardson, A. D. et al. Statistical properties of random CO2 flux measurement uncertainty inferred from model residuals. Agricultural and Forest Meteorology 148, 38–50 (2008).

Vuichard, N. & Papale, D. Filling the gaps in meteorological continuous data measured at FLUXNET sites with ERA-Interim reanalysis. Earth System Science Data 7, 157–171 (2015).
* * *

---

## Author Response (AR1)

**Dynamic INtegrated Gap-filling and partitioning for OzFlux (DINGO) response to reviewer comments**

Jason Beringer, Ian McHugh, Lindsay B. Hutley, Peter Isaac, and Natascha Kljun

As a whole the paper was well received but the reviewers and editor felt that the paper is rather technical with not strong scientific or methodological advancements (a lot of re-implementations of existing and published tools, not real validations). The editor also stated that at the same time the authors have an important paper in the context of the OzFlux Special Issue, because part of what this community did in the last years and for this reason the editor thinks that should be included. After discussion the editor felt that the paper should be re-organized as Technical note. We have still addressed the comments made by the reviewers that are relevant (see below).

In addition the editor also raised a specific question "(already raised by a reviewer but not yet clear): authors say that "'GPP is equated to zero at night but is not forced to zero during the day. Since GPP is the difference between measured net ecosystem exchange and estimate ecosystem respiration, it incorporates random error ……. While these estimates are therefore unphysical, the effect of their removal is to filter the positive domain of the random error distribution, thereby converting random error to systematic error.'". I completely agree with the explanation but forcing GPP to zero nighttime is something that hidden this effect of the random errors in the EC data. For this reason it would be probably better to leave the positive and negative GPP in the night to better help a user to understand that there is an uncertainty in the data (and have a quantification).". Although the random error at nighttime can be positive or negative the absolute value of GPP could also be significantly non-zero. This could occur at nighttime due to biases in the measurement of NEP and calculation of Re and therefore GPP would have similar biases that compound. So the absolute value of GPP at night includes a bias and random error. Since we know physiologically that GPP must be zero we feel that the most robust GPP at nightime would be to force it to zero". We add this discussion to the text.

**Anonymous Referee #1**

The manuscript is well written and describes a useful system to automate gapfilling. I do not find anything radically new in the approach. Each step in the process is based on well known, and widely used approaches to gapfilling. The novelty, perhaps, is in the automation of the process that offers new ease of use and the potential for widespread application. The test for such a new automated process is the applicability and its wide usability. Many flux sites have such a process for their own site, and none (or few) considered publishing it. As long as the process is specific to only and nothing more than OzFlux, I do not think it makes an interesting scientific paper, and should be published as an OzFlux report or the DINGO software manual. This could be easily remedied, and I propose that the authors will make the effort to generalize DINGO so that it could be applied by any flux site that desires to.

- The reviewer is correct in pointing out there are a few novel scientific outcomes from the paper. However, we feel that the paper makes an important contribution to the OzFlux special issue of which this paper is a part. It forms part of an important contribution that documents the OzFlux network in terms of the overall vision (special issue overview paper) through to the different processing methodologies that can be applied (OzFlux QC and DINGO). Therefore in the context of the special issue we feel this paper is well positioned and in this case. We recognise that the paper is more of a technical nature and as such it could be submitted as a technical note in the journal if this is possible.

Remembering in the context of OzFlux and the special issue that dingo is defined as the dyamic INtegrated Gap filling and partitioning for **Ozflux** (DINGO). The OzFlux QC paper in the special issue is in this identical situation, however, in that paper it was suggested to remove any performance analysis results.

You write in the abstract that "DINGO was developed for Australian data but the frame work is applicable to any flux data or regional network". However, it seems to be hardcoded for the Australian data sources and therefore will technically no run for other sites. As there is no novelty to the formulation, I see one of the big advantages in the automation of the work process and particularly in the data retrieval for the gap-filling, which is usually a cumbersome and laborious process. Currently, when gap-filling the forcing data the software only reads Australian met data and Australian high resolution meteorological reanalysis. Is it possible to allow a generic, user provided, table with data from n met stations nearby? Each user can prepare the data from their local met stations but the process of data selection and correlation with the flux-site met obs can be done by DINGO. Can DINGO be made to access the ECMWF reanalysis? It is coarser than the Australian one, but global and would allow users worldwide to have an alternative to local met stations.

- We agree that it would be a worthy goal to enable DINGO to be utilised globally but this is a massive amount of effort to develop and test this and would therefore be part of a new suite of tools (not DINGO any more). We think that just by enabling DINGO to be applied globally does not add extra scientific value (as suggested was needed by the reviewer), however, it would certainly make it more usable to the community. We suggest that is beyond the scope of the current paper but would be willing to work towards this in a new platform that could be developed with others in the flux net community.
- Thinking ahead to a possible global version of this tool in future work, ECMWF would be a great source of gridded climate data. This would need to be input in conjunction with local meteorology data from local weather stations. Most of the time in the Australian flux network the correlations between flux tower data are better with local weather stations and therefore this is the preferred source of data for gap filling. However, access to global weather station data may be rather difficult.

For precipitation, which is very poorly generalized from one met station to the other (and particularly the time of precipitation) I suggest that even for the OzFlux data processing there could be a great advantage in using satellite-derived precipitation from TRMM or the new GPM products.

- This is a really great suggestion from the reviewers and at the time of producing DINGO, TRMM was the main satellite precipitation product, however, it is not suitable for higher latitudes and does not cover all of Australia. We could consider including global precipitation data from GMP in a future global version of this platform.

Similarly, the soil moisture model can offer the option to be driven by MODIS LAI and ECMWF as an alternative to the strictly Australian data sources. That should be easy as MODIS is extracted anyhow for the albedo gapfilling.

- At present the soil moisture model is run off-line with AWAP gridded meteorological data. This is performed every 6- 12 months. The soil moisture model is not currently part of the code. However, there is a great opportunity to embed the soil moisture model into the code and therefore it could be run using this type of global data.

There is nothing new in incorporating footprint and estimating the uncertainty of the gap-filling method. In fact, it is irresponsible not to do so. These are features that you say will be available in V13, but I see no purpose in publishing an application to a well known method that is not done yet. However, you say that version 13 is due July 2016. This is a few weeks from now, and before the revision will be due. Can you make this paper about the complete and fully functional version 13?

- Version 13 of dingo is now complete so now update the manuscript accordingly and this strengthens the paper considerably as it includes the important uncertainty calculation contributions. We have added this and a figure showing its output.

Other comments: Why are you using different meteorology to drive the soil water model (meteorology from the Bureau of Meteorology's Australian Water Availability Project) than the station and other met data already processed and gapfilled for the station location, if station data is missing?

- Again, currently the soil moisture model is run off-line with AWAP gridded meteorological data. This is performed every 6- 12 months. The soil moisture model is not currently part of the code. In a new are version of the code we could embed the model into DINGO and therefore we could use the gap filled meteorological data to drive the model.

The u* filtering section should come before the ANN gap-filling section as u* filtering creates most of the gaps that then need to be filled.

- We suggest that it is not appropriate to produce a new platform that is globally applicable for this paper and is well outside the scope of the special issue. However, we think it is worthwhile discussing the potential challenges and opportunities for making the tool globally available and would add this to a future development section.
- One of the very novel features in DINGO is the uncertainty calculations in version 13 and we will include them in the new version of the manuscript. Most flux tower sites will produce some estimates of random and model uncertainty and these are usually reported separately. In DINGO v13 the uncertainties are calculated for random, model and ustar uncertainty and these uncertainties are combined to provide a total uncertainty for the site, which has not been done to date. We will include this to enhance the scientific value of the paper.
- We're not sure what the referee is referring to with respect to the u* filtering as section 4 is on friction velocity which precedes section 5 on gap filling. Also we clearly state "Once the data have been u* filtered, they are used to train an ANN (see Section 2.2) using nighttime data only with inputs of Sws, Ts, Ta and EVI as known drivers of ecosystem respiration (Migliavacca et al., 2010)".

**Anonymous Referee #2**

GENERAL COMMENTS

This manuscript describes a dynamic integrated gap filling and partitioning tool (DINGO) developed for standardized processing of OzFlux station data. The tool is designed to gap fill both meteorological variables and fluxes, besides it partitions the net flux into its components (Gross Primary Productivity and Ecosystem Respiration). Although it raises the important issue of a standardized post processing for meteorological and eddy flux data at network level, unfortunately the manuscript doesn't introduce important novelty aspects, and it sounds more like a technical note than a scientific contribution. It provides a very detailed description of the tool, but lacks of analysis and interpretation. Besides, most of the figures included in the manuscript are standard diagnostic or result plots provided by the tool itself and not the result of an in depth analysis of the tool behavior, performance or applicability. Some important aspects that should be faced and deepened are not included, as for example: the evaluation of the tool performances and their dependence on gap percentage, length and distribution, a comparison with other existing gap-filling and partitioning tools, an analysis of the effects of the gap filling on annual sums... Results of the uncertainty estimation would also enrich the work, together with some performance analysis at site/PFT level. A technical observation: u* filtering should be applied before gap filling since it introduces gaps in the time series. For

the above mentioned reasons, I do not think that the manuscript, in its present form, is suitable for publication in Biogesciences.

- Also see comment responses to referee one. In addition, we also recognise that the paper is more of a technical note and we leave it to the editors discretion as to whether the paper should be submitted as a technical note to Biogeosciences. However, as mentioned above the paper makes an important contribution to the OzFlux special issue of which this paper is a part. It forms part of an important contribution that documents the OzFlux network in terms of the overall vision (special issue overview paper) through to the different processing methodologies that can be applied (OzFlux QC and DINGO). The OzFlux QC is very similar in scope and technical detail to this paper. We agree that the performance of the tool is important and we are planning a separate manuscript to address this across both OzFluxQC and DINGO processing tools as well as comparison with other available platforms such as EdiProc.
    - We actually do perform u* filtering before gap filling, so we will clarify the wording in the text to make this apparent. We state that "Once the data have been u* filtered, they are used to train an ANN (see Section 2.2) using nighttime data only with inputs of Sws, Ts, Ta and EVI as known drivers of ecosystem respiration (Migliavacca et al., 2010)."

**Anonymous Referee #3**

The manuscript presents an overview of the software framework and methods for automated gap-filling and partitioning of OzFlux tower network data. Overall, the manuscript does a nice job presenting this overview with the right amount of detail and rationale, and is well-written. The gap-filling and partitioning methods employed are generally state-of-the-art community standards. That said, there is not much new information here. Perhaps the greatest novelty lies in the gap-filling of ancillary drivers, where to my knowledge the only other recent work on this topic is that of Vuichard and Papale (2015). Although that work is not referenced, the present manuscript expands upon it to incorporate data from nearby weather stations and satellite remote sensing observations, with additional soil- and radiation-tailored gap-filling methods.

I have two major recommendations: 1) The manuscript could be made more impactful by including an analysis of performance of the gap-filling and partitioning results across all or part of the OzFlux network. This would demonstrate the utility/flexibility of the framework, its ability to address the grand science challenges identified in the introduction, and highlight areas for continued improvement. 2) The paper would benefit from a discussion of how the methods employed in this paper conform to or push the envelope of current community practice. This would clearly communicate the significance of the software suite and its novel contributions.

- We thank the reviewers for their highly useful comments. We were not aware of the Vuichard and Papale (2015) reference so we thank you for that. We point to the comments in response to referee one and two. Given the comments made reviewer one and two we suggest that such a network wide analysis of the performance of gap filling tools is best suited to a separate paper that includes both of the major tools used in the OzFlux network (OzFluxQC and DINGO). That paper would also include some discussion of current community practice and how we can advance new or best practice.

Specific comments:

- The representation of meteorological quantities should make more effort to be consistent with community standards. I have found Reifsnyder et al. (1991) particularly useful for this purpose, presenting the symbols, units, and notation for use in the journal Agricultural and Forest Meteorology. Conforming to these where possible/practical will improve the readability and reach of the manuscript. Also, both Fre (pg. 3, line 31) and

ER (Fig. 1) are used in the manuscript to represent ecosystem respiration, and both Fc (pg. 6, line 14) and NEE (Fig. 1) are used to represent net ecosystem exchange of $CO_2$.

- Thanks again for the Reifsnyder et al. (1991) paper this will be a useful reference source in the future. Unfortunately the terminology this paper derives from the OzFlux terminology used in the initial processing in OzFlux QC and this has transferred to this paper to be consistent. The terminology is defined there and has been published in the OzFlux QC special issue paper so we will maintain a consistent convention. We will however ensure that the use of terms are consistent throughout the manuscript.

- Pg. 5, line 17-20 & Pg. 6, line 9-11: Is there a threshold correlation below which the data are not gap-filled? A discussion of this topic is warranted.

- There is currently no threshold, below which the data are not gap filled but this does raise a possible issue. We looked back through the correlation values for each of the meteorological variables across a range of sites and across them the 'best' correlation was never less than 0.5 (which is more than acceptable). The DINGO tool does take the three nearest weather stations as well as the gridded meteorology to do the correlations and so amongst all of those sources then ranks the 'best' correlations. As a result the 'best' correlation is always adequate. However, the 'worst' stations can have correlations of less than 0.1. We have added a description of this to the text.

- Pg. 5, line 17-20: Linear regression is a good start for using nearby station data for gap-filling. However, even from Fig. 2a (which looks to be a very clean example), the best correlation has a distinct non-linear component at the low end of values. Using this relationship to gap-fill the time series then extends the data into minimums not actually observed at the site, which has the potential to influence down-stream modeled physiological responses. Perhaps the manuscript could address this topic as a discussion point, and/or include a few different fitting functions in future iterations (maybe piecewise linear fits?).

- The reviewer raises a good point and one which we have previously considered. It would be easy to improve the statistical fit by increasing the complexity of the model from linear to piecewise linear to polynomial to machine learning. However, we feel that without any *a priori* reason for thinking that the relationship should be non-linear, then it is a slippery slope to go down to chase the best statistical fit particularly when this could result in non-physically realistic relationships between the two variables. We have added the following text "It would be easy to improve the statistical fit by increasing the complexity of the model from linear to piecewise linear to polynomial to machine learning. Vuichard and Papale (2015) have shown the limits of a linear regression correction method when applying a bias correction on meteorological fields for which the bias between flux tower and gridded meterology (ERA-I) data did not vary linearly. Nevertheless, we feel that without any a priori reason for thinking that the relationship should be non-linear, then it may be unwise to chase the best statistical fit particularly when this could result in non-physically realistic relationships between the two variables."

- Pg. 8, line 9-12: It would be informative to show some stats addressing how well this procedure replicates actual variation in solar radiation under cloudy conditions across the OzFlux network (opposed to using nearby BoM site data). What about using the diurnal average approach employed for estimating incoming longwave radiation under cloudy conditions (pg. 9, line 6-8)?

- The Bureau of Meteorology insolation data we used are daily estimates as described in the manuscript – there are few ground-based BOM monitoring sites with half-hourly measurements of insolation (less than 10 across Australia), and none within a reasonable distance of the flux towers in the OzFlux network. The methodology described in the paper is simply a means of first correcting the daily estimates of insolation using the tower observations (where available), then downscaling these to half-hourly values.

- The BOM satellite- and model-derived estimates compare very favourably with daily cumulative insolation measured at the sites ($r^2$ = 0.9+), and this could easily be included in the manuscript, but is generally discussed in the literature cited. Cloudiness is more problematic when downscaling from daily to (e.g.) half-hourly interval. The smooth insolation curve produced by the described algorithms is unlikely to reliably be realistic relative to ground-based measurements of insolation (which - depending on cloud type – would be much more temporally variable) simply because we have no information about temporal variations on this scale.

- We *could* instead use the approach taken with incoming longwave radiation, but if we averaged the diel cycle over multiple days with similar insolation, here we would approach a smooth curve anyway (and the remaining variation would be meaningless because the amount of cloud cover over short periods is inherently unpredictable, at least deterministically). Moreover, since a cloudy summer day may have daily insolation levels similar to a clear winter day, the ranking algorithm may group and average days from very different times of year. This would have the effect of smearing sunrise and sunset times undesirably, and generally reducing the accuracy of insolation estimates generally because of the differing path length (and corresponding extinction) of the solar beam through the atmosphere at different times of year.

- Pg. 10, line 4-5: How was the ANN hidden layer architecture of 24 and 16 nodes arrived at? Appropriate model complexity can significantly impact ANN model performance. Recent related works using ANNs for gap-filling have tested for ideal architectures (Papale and Valentini, 2003; Knox et al. 2015; Baldocchi et al., 2016).

- There is a compromise in the ANN model between statistical performance, training and over fitting. We simply used trial and error (through a systematic change in the range of parameters in the ANN model). Importantly we examined the performance of the neural network model across three different temporal ranges (diel, seasonal and annual). Many ANN combinations can achieve a good statistical performance across the mean range but be very poor at capturing the diurnal variation, for example. We have optimised the parameters to these three scales. Will have added this to the text.

- Pg. 10. ANN procedure: How are the data split into training and testing/validation sets?

- The data are split into 80% training and 20% testing. This has been added to the text.

- Pg. 12, line 12: "u* corrected output" implies that that data were somehow corrected for low u* conditions. I recommend rephrasing. - Pg. 12, line 16: Please clarify which Fre variable was used (Fre_NN or Fre_Con).

- What is meant is that simply, once the Fc timeseries has been filtered for low u* values it is gap filled using ANN Fre model and the resultant output is labelled as Fc_ustar. We have clarified this in the text.

- Pg. 12, line 16: What does "ve" mean? (as in +ve, -ve)

- Positive and negative. We have expanded this in the text.

- Pg. 12, line 17-22: This explanation is confusing. Recommend rephrasing. Consider explaining that forcing GPP to zero at night removes positive and negative random error equally, but forcing any positive GPP values to zero during daytime would bias results because only negative random errors would remain. Also, I am not sure what is meant by "no random error in the Re ANN calculation", since random error is relevant in any model fit to real data.

- Will have amended to read: 'GPP is equated to zero at night but is not forced to zero during the day. Since GPP is the difference between measured net ecosystem exchange and estimate ecosystem respiration, it incorporates random error that is superimposed on the measurements (and potentially

also systematic error in the model), and may be correspondingly higher or lower than 'true' GPP. As a result, some GPP estimates may switch to positive sign when the signal:noise ratio is low (e.g. early morning / later afternoon). While these estimates are therefore unphysical, the effect of their removal is to filter the positive domain of the random error distribution, thereby converting random error to systematic error.'

- Pg. 14, line 16-27: Since the ANN is employed for gap-filling, why not use the model residuals as an estimate of the random error? The model residuals of high performance gap-filling algorithms such as the ANN provide a good, if not conservative estimate of the random uncertainty (Moffat et al. 2007, Richardson et al. 2008). The daily differencing approach is muc more conservative as it includes natural environmental variability as a result of variation in the flux footprint. This would help alleviate some of the double-counting of uncertainty mentioned in a later paragraph (pg. 15, line 7-12).

- Explicitly accounting for and interpreting the effects of radial variations in the flux source area requires relatively complex site-specific analysis that is presently considered to be beyond the scope of the development of this tool. Thus we would prefer to use a more conservative approach to uncertainty analysis that *does* include such effects, though we could be explicit in our reasoning in the manuscript. The major issue with double-counting across methods is that the model uncertainty is based on comparing observations to model estimates. The observations contain random error but we are counting model error as any discrepancy between model and measurement, thereby implicitly assuming a pure signal. Switching to characterisation of random error using model residuals doesn't solve this problem, because in the case of model error, the random error is present but not explicitly characterised.

- There are arguments for alternative methods of calculating model error. For example, we have explored repeatedly degrading an ANN-produce NEE signal with random error and training and deploying a model based on this data. However, this accounts only for the effects of random error on model parameterisation, thereby excluding any systematic error (for example associated with missing drivers) from model uncertainty estimates. Thus we consider our approach – which implicitly includes the effects of any systematic error – defensible.

- We can add discussion to the text if required?

- Pg. 14, line 19: Please indicate what lowercase delta means.

- Will be amended.

- Pg. 14, line 18-21: Why must random error be calculated over bins of u*? (rather than over bins of (or a regression with) flux magnitude) –

- Will be amended to read: 'Since random error is heteroschedastic, its magnitude (expressed by the standard deviation or $\sigma[\delta]$, since the mean is expected to be zero) must be expressed as a function of the mean flux magnitude.'

Section 2.6 Uncertainties: *This section would benefit from a discussion on the uncertainty propagated along the entire* from a discussion on the uncertainty propagated along the entire processing chain described in the manuscript (i.e. using gap-filled ancillary drivers value to gap-fill fluxes).

- Agreed. This is an area of expansion of the uncertainty estimation planned for the future.

- Pg. 17, line 16-18: What is the suggested turnaround time from data collection to quality output from the procedure outlined in the manuscript?

- Dingo will process the level 3 data from the OzFlux QC through to fully gap filled and partitioned data in less than 30 minutes on a desktop PC. When the Barr et al. Methodology is used for computing u* this procedure adds another hour or so depending on the length of the dataset and the number of iterations in the bootstrapping. In addition, the uncertainty code adds another hour or so depending on the number of iterations chosen and the length of the dataset. This can be added to the manuscript.

Technical corrections

- All bulleted lists: In some cases the lack of sentence case following the bullet point makes sense. However, in many cases the sentence following the bullet is a standalone sentence and should use sentence case.

- Have corrected this.

- Pg. 10, line 29: "stable conditions" should specifically reference "atmospherically stable conditions"

- Have changed to atmospherically stable conditions.

- Pg. 13, line 26: I am confused by the sentence fragment "Once complete daily time series for the parameter estimates,. . .". What about this: "Once daily estimates for the parameters are generated,. . ."

- The referees suggestion is clearer and have amended this in the text.

- Labels in figures need cleaning up (e.g. Fig. 4 y-axis – what is CABLE?, Fig. 5 y-axis – make this human readable, Fig. 9b legend – what is the difference between Fc and Fc?, use superscript and subscripts where appropriate)

- We have cleaned up the figures to have appropriate units. We have removed the titles from the plots in each figure. CABLE is the other name for the BIOS model used for soil gap filling but we have removed this from the figure.
- In the legend of figure 9 the two colours are as stated in the caption "weekly plot with period of more than 30% missing data shown in light blue". We will tidy this up and make the legend clear.

- The manuscript needs thorough editing for typographical errors

- Will do

References cited

Baldocchi, D. et al. The impact of expanding flooded land area on the annual evaporation of rice. Agricultural and Forest Meteorology 223, 181–193 (2016).

Knox, S. H. et al. Agricultural peatland restoration: effects of land-use change on greenhouse gas ($CO_2$ and $CH_4$) fluxes in the Sacramento-San Joaquin Delta. Global Change Biology 21, 750–765 (2015).

Moffat, A. M. et al. Comprehensive comparison of gap-filling techniques for eddy covariance net carbon fluxes. Agric. For. Meteorol. 147, 209–232 (2007). Papale, D. & Valentini, A. A new assessment of European forests carbon exchanges by eddy fluxes and artificial neural network spatialization. Global Change Biology 9, 525–535 (2003).

Reifsnyder, W. E., McNaughton, K. G. & Milford, J. R. Symbols, units, notation. A statement of journal policy. Agricultural and Forest Meteorology 54, 389–397 (1991).

Richardson, A. D. et al. Statistical properties of random $CO_2$ flux measurement uncertainty inferred from model residuals. Agricultural and Forest Meteorology 148, 38–50 (2008).

Vuichard, N. & Papale, D. Filling the gaps in meteorological continuous data measured at FLUXNET sites with ERA-Interim reanalysis. Earth System Science Data 7, 157– 171 (2015).

[revised manuscript text omitted]

20   of AWS data is as follows:

[revised manuscript text omitted]

5. the code module also outputs many diagnostic plots including general model performance (Fig. Fig. S6a), monthly time series (Fig. Fig. S6b), a 30 minute timeseries data check (Fig. Fig. S6d) and a check to see if the ANN performs diurnally (Fig. Fig. S6c).

**S12.4 Friction velocity (u*) filtering**

The eddy covariance technique is well known to underestimate turbulence fluxes of carbon dioxide under atmospherically stable conditions, particularly at night time where the surface can be decoupled from the measurements at a height above the canopy (Goulden et al., 1996). An excellent overview of this subject is given by Aubinet et al. (2012). This problem has been
5   shown to impact fluxes across some Australian sites such as the old growth Mountain ash site (Kilinc et al., 2010) and a cool temperate eucalypt forest (van Gorsel et al., 2009). The primary technique to deal with this is to exclude data taken where the eddy covariance measurements is not representative of the true flux. Typically this occurs when u* values are below a critical threshold (Goulden et al., 1996). There are several ways to calculate the friction velocity ($u^*$) threshold as shown in Aubinet et al. (2012) and in DINGO we calculate the threshold based on both the procedures of (1) Reichstein et al. (Reichstein et al.,
10  2005a) and (2) Barr et al. (2013). Alternatively the user may select their own constant value. The threshold used for subsequent filtering is user selectable but the threshold determined using Barr et al. (2013) is used by default. Whatever choice is made the resulting u* that is used is saved to the main file named 'ustar_used'. The two methods are implemented as follows:

1. For for the Reichstein et al. (Reichstein et al., 2005a) approach, the non-gap-filled data set is split into 6 temperature classes of the same sample size (according to quantiles) and for each temperature class the set is split into 20 u*-
15     classes according to Papale et al., (2006). The threshold is defined as the u*-class where the night-time flux reaches more than 95% of the average flux at the higher u*-classes. The threshold is only accepted if the temperature class and u* are not or only weakly correlated ($|r| < 0.3$). The final threshold is defined as the median of the thresholds of the (up-to) six temperature classes. This procedure is applied to the entire dataset, giving a maximum, but conservative u* threshold (Fig. Fig. S7). The maximum value is saved as 'ustar_Reich_max'. In addition, the u* threshold is
20     calculated continuously using a 1 month moving window to account for seasonal variation of vegetation structure (Fig. Fig. S7) and saved as 'ustar_Reich_var', and.

2. The the Barr et al. (2013) approach uses a change point detection technique to objectively identify the best estimate and uncertainty range for the u* threshold. In brief, the method involves fitting a two-phase linear regression model to all possible change points within a nocturnal data sample (i.e. $2 \le c \le n-1$), finding the change point that minimised
25     the sum of squared error and establishing whether its performance was statistically significantly improved relative to a reduced form (no change point) null model. For each year, the data were divided into sequential samples of $n = 10^3$, with 50% overlap between samples. Each sample was in turn divided into four temperature classes, ordered by $u_*$ then bin-averaged (n = 5 for each bin) to reduce the effects of random error. To increase the sample size, the data were bootstrapped ($n = 10^3$) by simply randomly sampling (with replacement) records from the original dataset and
30     rerunning the analysis. This yielded a (Gaussian) distribution of u* thresholds, the mean and 95% confidence interval of which were taken as the best estimate and uncertainty of the u* threshold. As per Barr et al. (2013) we identified the dominant mode of the NEE dependency on u* (i.e. positive or negative slope below the change point), and rejected all thresholds from the non-dominant mode (in practice the negative dependency slope was very rare), and rejected

any annual analysis where the number of valid change points (across all temperature strata and bootstraps) was less than 4000 or less than 20% of the total.  Annual u* statistics are saved and the annual Barr u* threshold is written to the main file as 'ustar_Barr'.

**S12.5 Calculation of Fre and GPP**

**S12.5.1 ANN modelling of Fre**

Once the data have been u* filtered, they are used to train an ANN (see Section 2.2) using nighttime data only with inputs of Sws, Ts, Ta and EVI as known drivers of ecosystem respiration (Migliavacca et al., 2010).  Importantly, in DINGO we also only use flux data from the first 3 hours after sunset where the canopy is still coupled with the atmosphere, as shown in Van Gorsel et al. (2007).  This makes the selection of data for the ANN model more conservative than using the entire nighttime period.  This option is also user selectable.  Nighttime, daytime and evening periods (first 3 hours after sunset) are determined from sunset and sunrise times calculated precisely from solar geometry accounting for elevation, latitude, longitude and day of year and saved as an integer 'day_night' (day=1, evening=2, night=3).

Once the Fc timeseries has been filtered for low u* values it is gap filled using ANN Fre model and the resultant output is labelled as Initially, the Fc timeseries has nighttime gaps from the u* threshold filtering and these are filled using the ANN Fre model with the u* corrected output labelled as 'Fc_ustar'.  Next, the ANN model output trained using nightime data is then extrapolated to the daytime data to create a continuous timeseries of modelled Fre (Fre_NN) that is used to fill missing gaps in the nighttime data where data where either filtered out using the u* threshold or where missing.  This creates a continuous time series of Fre (Fre_Con) that is a combination of valid observations and ANN model output.  GPP is then calculated as the difference between Fc and Fre (GPP_Con) where our sign convention is -venegative is a net flux into ecosystem and +vepositive away from it.  GPP is equated to zero at nighttime but it is not forced to zero at any other time.  This is because for any 30 minute value, the random error in Fc results in values of Fc being higher or lower than the true value.  Due to this error, it is possible then to get values of GPP that have either a higher or lower random error component.  GPP is equated to zero at night but is not forced to zero during the day. Since GPP is the difference between measured net ecosystem exchange and estimate ecosystem respiration, it incorporates random error that is superimposed on the measurements (and potentially also systematic error in the model), and may be correspondingly higher or lower than 'true' GPP. As a result, some GPP estimates may switch to positive sign when the signal:noise ratio is low (e.g. early morning / later afternoon). While these estimates are therefore unphysical, the effect of their removal is to filter the positive domain of the random error distribution, thereby converting random error to systematic error.There is also no random error in the Re ANN calculation and if we force these GPP values to zero then there is a bias in the GPP integrals (e.g. daily totals) because we only remove the values that have a negative error. Forcing GPP to zero during the daytime leads to ever so slightly higher GPP

because lower values are cut off particularly in the evening and morning when GPP approaches zero. We feel that it is not appropriate to leave the positive and negative GPP in the night because although the random error at nighttime can be positive or negative the absolute value of GPP could also be significantly non-zero. This could occur at nighttime due to biases in the measurement of Fc and calculation of Fre and therefore GPP would have similar biases that compound. So the absolute value of GPP at night includes a bias and random error and since we know physiologically that GPP must be zero we feel that the most robust GPP at nightime is to force it to zero.

[revised manuscript text omitted]

Reichstein, M., Falge, E., Baldocchi, D., Papale, D., Aubinet, M., Berbigier, P., Bernhofer, C., Buchmann, N., Gilmanov, T., Granier, A., Grunwald, T., Havrankova, K., Ilvesniemi, H., Janous, D., Knohl, A., Laurila, T., Lohila, A., Loustau, D., Matteucci, G., Meyers, T., Miglietta, F., Ourcival, J.-M., Pumpanen, J., Rambal, S., Rotenberg, E., Sanz, M., Tenhunen, J.,

25  Seufert, G., Vaccari, F., Vesala, T., Yakir, D. and Valentini, R.: On the separation of net ecosystem exchange into assimilation and ecosystem respiration: review and improved algorithm, Glob. Chang. Biol., 11(9), 1424–1439, doi:10.1111/j.1365-2486.2005.001002.x, 2005a.

Reichstein, M., Katterer, T., Andren, O., Ciais, P., Schulze, E. D., Cramer, W., Papale, D. and Valentini, R.: Temperature sensitivity of decomposition in relation to soil organic matter pools: critique and outlook, Biogeosciences, 2(4), 317–321,

30  2005b.

Richardson, A. D. and Hollinger, D. Y.: Statistical modeling of ecosystem respiration using eddy covariance data: Maximum likelihood parameter estimation, and Monte Carlo simulation of model and parameter uncertainty, applied to three simple models, Agric. For. Meteorol., 131(3–4), 191–208, doi:10.1016/j.agrformet.2005.05.008, 2005.

Ruimy, A., Jarvis, P. G., Baldocchi, D. D. and Saugier, B.: CO2 Fluxes over Plant Canopies and Solar Radiation: A Review,

Elsevier., 1995.

Sanders, A. F. J., Verstraeten, W. W., Kooreman, M. L., Leth, T. C. van, Beringer, J. and Joiner, J.: Spaceborne sun-induced vegetation fluorescence time series from 2007 to 2015 evaluated with Australian fluxtower measurements, Remote Sens., Accepted 1, 2016.

5 Savitzky, A. and Golay, M. J. E.: Smoothing and Differentiation of Data by Simplified Least Squares Procedures., Anal. Chem., 36(8), 1627–1639, doi:10.1021/ac60214a047, 1964.

Twine, T. E. and Kucharik, C. J.: Evaluating a terrestrial ecosystem model with satellite information of greenness - art. no. G03027, J. Geophys. Res., 113(G3), 3027, 2008.

Vuichard, N. and Papale, D.: Filling the gaps in meteorological continuous data measured at FLUXNET sites with ERA-

10 Interim reanalysis, Earth Syst. Sci. Data, 7(2), 157–171, doi:10.5194/essd-7-157-2015, 2015.

[revised manuscript text omitted]

**Figure 8S9: DINGO produces many diagnostic plots including energy balance closure (difference between Fh+fe and Fn-Fg) for a) all hours of 30 minute values  b) daytime hours of 30 minute values c) nightime hours of 30 minute values d) daily means. Fh is sensible heat flux (W m⁻²), Fe is latent heat flux (W m⁻²), Fn is net radiation (W m⁻²) and Fg is ground heat flux (W m⁻²). Example is from the Calperum flux tower (see Beringer et al.  (2016) for details) for 2012.**

[Figure]

**Figure 9S10:** DINGO produces many diagnostic plots for all fluxes and meteorological variables (in this case net ecosystem exchange (Fc – $\mu$mol m$^{-2}$ s$^{-1}$)) including a) amount of missing data and data not gap-filled (percent) and b) weekly plot (dark-blue) with periods of more than 30% missing data shown in light blue. Example is from the Calperum flux tower (see Beringer et al. (2016) for details) for 2010 to 2013.

[Figure]

**Figure 10S11: Example of summary fingerprint plot of a) net ecosystem exchange (Fc), b) ecosystem respiration (Fre) and c) gross primary production (GPP) all in units of μmol m⁻² s⁻¹. Example is from the Calperum flux tower (see Beringer et al. (2016) for details) for 2010 to 2013.**

[Figure]

**Figure S12: Summary timeseries plot of daily means and a 30 day running mean (dark line) of net ecosystem exchange (Fc, yellow), ecosystem respiration (Fre, red) and c) gross primary production (GPP, green) all in units of μmol m⁻² s⁻¹. Example is from the Calperum flux tower (see Beringer et al.  (2016) for details) for 2010 to 2014.**

[Figure]

**Figure S13: Example of results plot of cumulative carbon uptake (t C ha$^{-1}$ yr$^{-1}$) (negative being uptake) by year. Example is from the Calperum flux tower (see Beringer et al. (2016) for details) for 2010 to 2014 that in this case shows the system swings from being a sink to source from one year to the next.**

---

## Author Response (AR2)

**Dynamic INtegrated Gap-filling and partitioning for OzFlux (DINGO) response to reviewer comments**

Jason Beringer, Ian McHugh, Lindsay B. Hutley, Peter Isaac, and Natascha Kljun

The associate editor has given us food for thought and have asked us to consider his comment below.

While I agree that it includes both random and systematic I think that forcing to zero is an error because one one side you remove this systematic error only nighttime (daytime is still there and you don't see it) but also because you mask a problem and for a user this is negative. If I get a DINGO result where a site has total nighttime GPP of 5 gC or -5 gC I will get the important information that there could be a bias in the data. If you force to 0 this info is lost…

As it turns out the way that DINGO (and OzFlux QC) works the value of GPP naturally equates to zero at night time anyway and therefore there is no need to force to zero.  So we have rewritten the paragraph as follows to explain that point:

"Once the Fc timeseries has been filtered for low u* values it is gap filled using ANN Fre model and the resultant output is labelled as 'Fc_ustar'.  Next, the ANN model output trained using nightime data is then extrapolated to the daytime data to create a continuous timeseries of modelled Fre (Fre_NN) that is used to fill missing gaps in the nighttime data where data where either filtered out using the u* threshold or where missing.  This creates a continuous time series of Fre (Fre_Con) that is a combination of valid observations and ANN model output.  GPP is then calculated as the difference between Fc and Fre (GPP_Con) where our sign convention is negative is a net flux into ecosystem and positive away from it.  GPP naturally equates to zero numerically at nighttime.  This is because we use the u* filter to exclude low turbulence conditions at night and then assume that the remaining observations of Fc are valid measurements of Fre.  Hence, when u* is above the threshold, Fc and Fre have the same values and since GPP = -Fc + Fre, this gives GPP = 0.  In addition, using the modelled Fre from the ANN we predict Fre for those times at night when Fc is missing (through QA/QC or rejection by u* filter) and for the daytime.  At night, when u* is below the threshold the ANN prediction replaces Fre and Fc (same value) and since again, GPP = -Fc + Fre, this gives GPP = 0.  GPP is not forced to zero during the day and this can sometimes result in GPP being positive (biologically nonsense) particularly close to sunrise and sunset. Since GPP is the difference between measured Fc and estimated Fre, it incorporates random error that is superimposed on the measurements (and potentially also systematic error in the model), and may therefore be correspondingly higher or lower than the 'true' value. As a result, some GPP estimates may switch to positive sign when the signal:noise ratio is low (e.g. early morning / later afternoon). While these estimates are therefore unphysical, the effect of their removal is to filter the positive domain of the random error distribution, thereby converting random error to systematic error that leads to slightly higher GPP and therefore we do not force GPP to zero during the daytime."